# Coordinated molecular and ecological adaptations underlie a highly successful parasitoid

Lan Pang[1†], Gangqi Fang[2,3†], Zhiguo Liu[1], Zhi Dong[1], Jiani Chen[1], Ting Feng[1,4], Qichao Zhang[1,4], Yifeng Sheng[1,4], Yueqi Lu[1,4], Ying Wang[1,4], Yixiang Zhang[2,3], Guiyun Li[2], Xuexin Chen[1,4,5], Shuai Zhan[2,3]*, Jianhua Huang[1,4]*

[1]Institute of Insect Sciences, Ministry of Agriculture Key Lab of Molecular Biology of Crop Pathogens and Insect Pests, College of Agriculture and Biotechnology, Zhejiang University, Hangzhou, China; [2]CAS Key Laboratory of Insect Developmental and Evolutionary Biology, CAS Center for Excellence in Molecular Plant Sciences, Chinese Academy of Sciences, Shanghai, China; [3]CAS Center for Excellence in Biotic Interactions, University of Chinese Academy of Sciences, Beijing, China; [4]Key Laboratory of Biology of Crop Pathogens and Insects of Zhejiang Province, Zhejiang University, Hangzhou, China; [5]State Key Lab of Rice Biology, Zhejiang University, Hangzhou, China

*For correspondence:
szhan@sibs.ac.cn (SZ);
jhhuang@zju.edu.cn (JH)

[†]These authors contributed equally

**Abstract** The success of an organism depends on the molecular and ecological adaptations that promote its beneficial fitness. Parasitoids are valuable biocontrol agents for successfully managing agricultural pests, and they have evolved diversified strategies to adapt to both the physiological condition of hosts and the competition of other parasitoids. Here, we deconstructed the parasitic strategies in a highly successful parasitoid, *Trichopria drosophilae*, which parasitizes a broad range of *Drosophila* hosts, including the globally invasive species *D. suzukii*. We found that *T. drosophilae* had developed specialized venom proteins that arrest host development to obtain more nutrients via secreting tissue inhibitors of metalloproteinases (TIMPs), as well as a unique type of cell—teratocytes—that digest host tissues for feeding by releasing trypsin proteins. In addition to the molecular adaptations that optimize nutritional uptake, this pupal parasitoid has evolved ecologically adaptive strategies including the conditional tolerance of intraspecific competition to enhance parasitic success in older hosts and the obligate avoidance of interspecific competition with larval parasitoids. Our study not only demystifies how parasitoids weaponize themselves to colonize formidable hosts but also provided empirical evidence of the intricate coordination between the molecular and ecological adaptations that drive evolutionary success.

## eLife assessment

The paper presents **valuable** insights into the success of the parasitoid *Trichopria drosophilae* on *Drosophila suzukii*, elucidating the importance of both molecular adaptations, such as specialized venom proteins and unique cell types, ecological strategies, including tolerance of intraspecific competition and avoidance of interspecific competition. Through **convincing** methodological approaches, the authors demonstrate how these adaptations optimize nutrient uptake and enhance parasitic success, highlighting the intricate coordination between molecular and ecological factors in driving parasitization success.

## Introduction

It is estimated that approximately 10% of insect species are parasitoids, with females typically depositing their eggs either in (endoparasitoids) or on (ectoparasitoids) the bodies of their hosts, leading to the consumption of the host by the parasitoid's developing offspring (*Eggleton and Belshaw, 1992*; *Vinson and Iwantsch, 1980*; *Charnov, 1994*; *Harvey and Strand, 2002*; *Pennacchio and Strand, 2006*; *van de Kamp et al., 2018*). The antagonistic coevolution between parasitoids and hosts has spurred a vast array of adaptive and counter-adaptive strategies, in which the hosts have developed better immune responses and/or evasion behaviours to prevent parasitism; however, parasitoids have evolved diversified parasitic strategies to enable successful parasitism (*Müller and Schmid-Hempel, 1993*; *Kacsoh et al., 2013*; *Huang et al., 2021*; *Chen et al., 2021*; *Pang et al., 2022*; *Wertheim, 2022*). This evolutionary arms race fosters a dynamic ecosystem characterized by either the emergence of strong resistance mechanisms in certain host species or the evolution of powerful parasitic strategies in certain parasitoids (*Beckage and Gelman, 2004*; *Colinet et al., 2007*; *Wang et al., 2018a*; *Gasmi et al., 2021*).

One example notable for strong resistance to parasitization is the spotted-wing fruit fly *Drosophila suzukii* (Diptera: Drosophilidae). This invasive species originated in Asia and has spread rapidly throughout the world over the last decade, with records in Europe (since 2008), North America (since 2008), South America (since 2013), and Africa (since 2017) (*Deprá et al., 2014*; *Dos Santos et al., 2017*; *Boughdad et al., 2021*). *D. suzukii* attacks a broad spectrum of soft-skinned fruits and causes serious economic damage to fruit and wine production (*Rota-Stabelli et al., 2013*; *Asplen et al., 2015*; *Knapp et al., 2021*). Despite the broad interest in seeking natural enemies, the most well-recognized drosophilid parasitoids cannot parasitize *D. suzukii* (*Chabert et al., 2012*; *Poyet et al., 2013*; *Lee et al., 2019*).

We conducted field investigations to identify natural parasitoids of *D. suzukii* from 2016–2020. We placed traps containing the favourite fruits of *D. suzukii* (e.g. banana pieces, cherries, and grapes) on *Myrica rubra* trees at several locations in East China (e.g. Hangzhou, Ningbo, and Taizhou), where *D. suzukii* is heavily distributed (*Wang et al., 2020*; *Yu et al., 2021*). Among the reclaimed parasitic wasps, we primarily characterized two species: *Trichopria drosophilae* (Td) (Hymenoptera: Diapriidae), which parasitizes *D. suzukii* pupae, and *Asobara japonica* (Aj) (Braconidae), which parasitizes larvae. Remarkably, we observed a constant proportion of Td in repeated traps, which attracted the most interest. The parasitic success of Td on *D. suzukii* has been documented in the USA and Europe independently (*Knoll et al., 2017*; *Wang et al., 2018b*; *Yi et al., 2020*). Thus, Td provides a valuable model for studying the parasitic success in a pupal parasitoid and serves as a promising control agent for invasive pests. However, the mechanisms underlying the successful parasitization of Td remains largely unknown.

By conducting an interdisciplinary study that integrates multi-omics analyses and functional behavioural assays, we found that, to adapt to the limited resources of the pupal host, this parasitoid has evolved weaponized venom and teratocyte cells to help arrest host development and speed up the digestion of host tissues. In addition, Td allowed intraspecific competition as a conditional collaboration to compensate for its inability to find more appropriate hosts (the younger ones) but effectively avoided interspecific competition with larval parasitoids. The availability of multi-omics resources and knowledge of parasitism will further benefit the optimization and utilization of Td in pest management.

## Results

### Td effectively parasitizes *D. suzukii* and a broad range of drosophilids

Both Td and Aj were trapped in the field and maintained in the laboratory for further study. Aj wasps mainly parasitize 2nd instar *Drosophila* larvae, showing a 60% parasitism rate (the proportion of hosts parasitized) and a 48% emergence rate (the proportion of hosts with wasps hatched) on *D. suzukii*. In comparison to Aj, Td laid eggs in the pupae (*Figure 1A*) and presented a much better parasitic performance on *D. suzukii*, showing an 85% parasitism rate (p<0.001) and a 77% emergence rate (p<0.001) (*Figure 1B and C*). To investigate the host range, we tested their parasitic efficiencies in another five *Drosophila* species, including three species in the *melanogaster* subgroup (*D. melanogaster*, *D. simulans*, and *D. santomea*) and two species outside the *melanogaster* subgroup (*D. pseudoobscura*

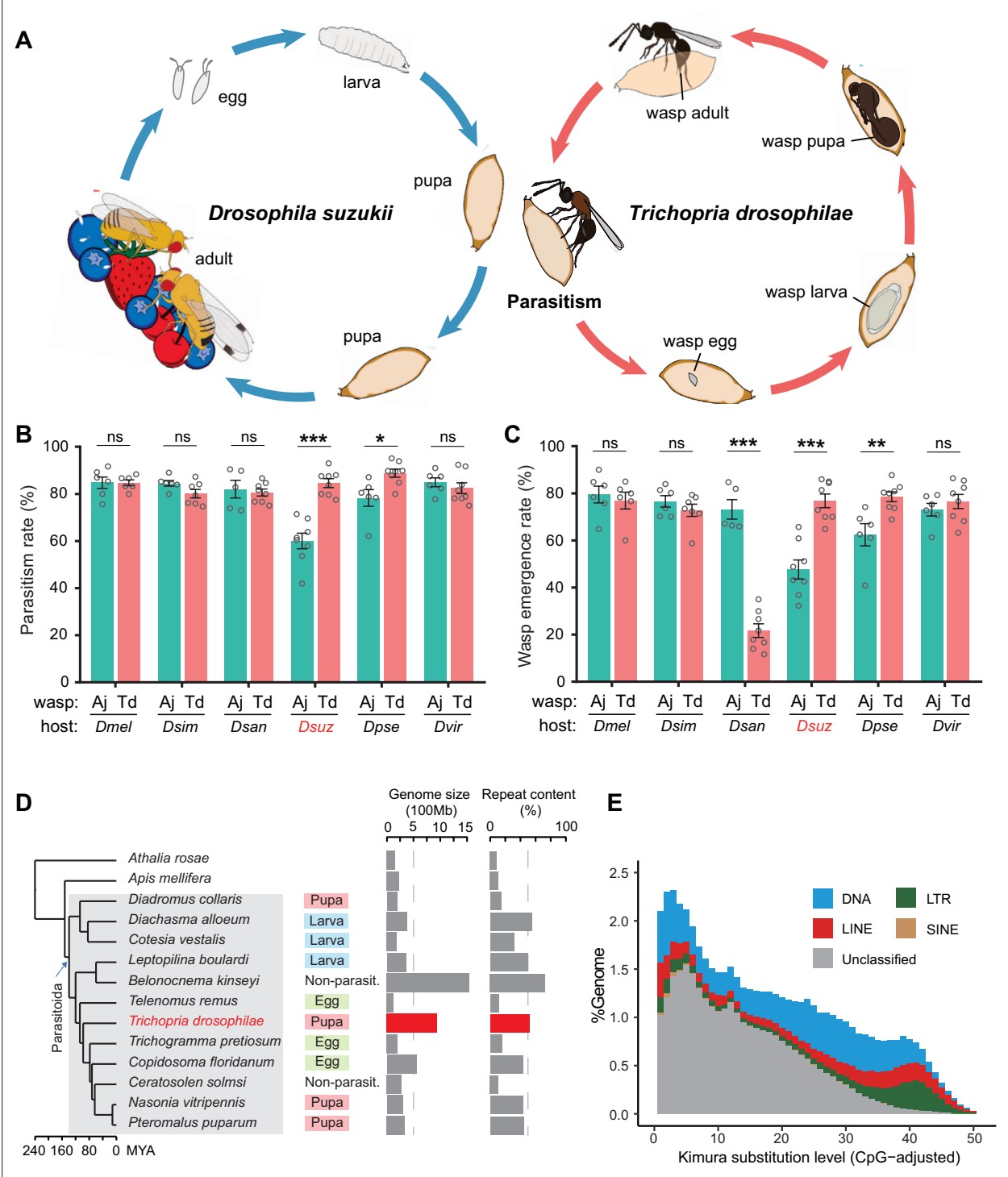

**Figure 1.** Chromosome-level assembly of a generalist drosophilid parasitoid genome. (**A**) The life cycle of *T. drosophilae* (parasitoid) and *Drosophila* (host). *T. drosophila* (Td) is a pupal parasitoid that deposits eggs into *Drosophila* host pupae. As a generalist parasitoid, Td can successfully parasitize a variety of *Drosophila* species, including the well-known invasive pest *D. suzukii*. (**B**) and (**C**) The comparison of parasitization performance between Td and *A. japonica* (Aj). The parasitism rates (**B**) and wasp emergence rates (**C**) in six *Drosophila* species parasitized by Aj and Td. At least five biological replicates were performed. Data represent the mean ± SEM. Significance was analysed by two-way ANOVA with Sidak's multiple comparisons test (ns, not significant; *p<0.05; **p<0.01; ***p<0.001). *Dmel, D. melanogaster; Dsim, D. simulans; Dsan, D. santomea; Dsuz, D. suzukii; Dpse, D. pseudoobscura; Dvir, D. virilis.* (**D**) Genome size and repeat content of representative parasitoid species in the context of phylogeny. Parasitoid species were selected based on the presence of high-quality genome reference and the representativeness of the group. The corresponding host life stage is

*Figure 1 continued on next page*

*Figure 1 continued*

shown in the coloured box. (**E**) The age and relative abundance of repeat element classes in Td genome. Kimura divergence from the consensus was estimated to indicate the burst time of repeats. See *Figure 1—figure supplement 2* for the plots of all involved parasitoid species.

The online version of this article includes the following source data and figure supplement(s) for figure 1:

**Source data 1.** Related to *Figure 1B*.

**Source data 2.** Related to *Figure 1C*.

**Figure supplement 1.** Chromosome features of *Trichopria drosophilae* (Td).

**Figure supplement 2.** Distribution of repeat content among representative parasitoid species.

**Figure supplement 3.** Phylogeny and ortholog content across representative parasitoid species.

and *D. virilis*). Of these hosts, Td showed high parasitism rates in all species (ranging from 80 to 89%) and high emergence rates in most species (ranging from 73 to 79%), except in *D. santomea* (22%) (*Figure 1B and C*). Aj could also parasitize these drosophilid species but showed much poorer performance in the non-melanogaster subgroup (*Figure 1B and C*). The parasitism spectrum in the laboratory showed that Td is also a generalist parasitoid with a wide host range, including *D. suzukii* and many other *Drosophila* species.

## Td shows a relatively large genome and encodes many copies of tissue inhibitors of metalloproteinase (*Timp*) genes

Genomic information is critical to characterize the molecular bases underlying parasitic strategies, which is absent for Td. We first sequenced and de novo assembled Td reference genome based on PacBio long reads (120.1 Gb). The assembled 938.6 Mb genome consisted of 1633 contigs with an N50 size of 146.9 Mb (*Supplementary file 1*). BUSCO estimation showed a high level of completeness (96%) in terms of insect proteins (*Supplementary file 1*). By combining 143.7 Gb Hi-C data, 97.8% of assembled contigs were further anchored to six pseudochromosomes, agreeing with the karyotype staining estimation (*Figure 1—figure supplement 1*).

The assembled Td genome (938.6 Mb) represents an occasionally large one in parasitoids, since most published parasitoid and hymenopteran genomes were relatively small (<500 Mb) (*Figure 1D*). Parasitic lifestyle was previously assumed to be associated with reduced genome size and complexity (*Moran, 2002*; *Wicke et al., 2013*; *Poulin and Randhawa, 2015*). While this study was in preparation, Ye et al. reported two large genomes (~950 Mb) of *Anastatus* parasitoids (Chalcidoidae) and proposed that the unexpected genome expansion was due to recent bursts of long terminal repeat (LTR) retrotransposons. Similarly, we found recent bursts of transposon elements (TEs) in Td genome (*Figure 1E*), although a different class of TEs (DNA transposons, rather than LTRs) was the most abundant (*Figure 1—figure supplement 2*). We further found that the age of the TE burst and the abundance of different classes both highly varied across parasitoid genomes, regardless of the whole genome size (*Figure 1D*, *Figure 1—figure supplement 2*). The previous study has also proposed that the expanded genome of *Anastatus* is stabilized owing to the expansion of the *Piwi* gene family (*Ye et al., 2022*); however, Td genome only encodes three *Piwi* genes, which is similar to most other Hymenoptera species but much lower than those of *Anastatus* (16 and 30 genes). These patterns suggest that the mechanisms to maintain the stability of genome size may vary between species and that the highly dynamic evolution of repeat contents has shaped great diversity across parasitoid genomes.

By combining *ab initio* gene signatures, transcriptome evidence, and homologue alignments, we generated an official gene set consisting of 16,287 protein-coding genes for Td (*Supplementary file 1*; Methods). We used one-to-one orthologous protein genes to infer phylogenetic relationships across representative parasitoids. In Hymenoptera, parasitoidism evolved and underwent massive habitat diversification, including scattered secondary losses (*Figure 1D*; *Peters et al., 2017*; *Blaimer et al., 2023*; *Polaszek and Vilhemsen, 2023*). Adaptations to a certain host life stage, i.e., eggs, larvae, or pupae, evolved independently in different sublineages (*Figure 1D*). Td was placed as a sister sublineage to a large group of chalcid parasitoids with a divergence time of approximately 98 million years ago (Mya) (*Figure 1D*). Its divergence from *Leptopilina*, another major genus of drosophilid parasitoids (Cynipoidea), dates back to 120 Mya (*Figure 1D*).

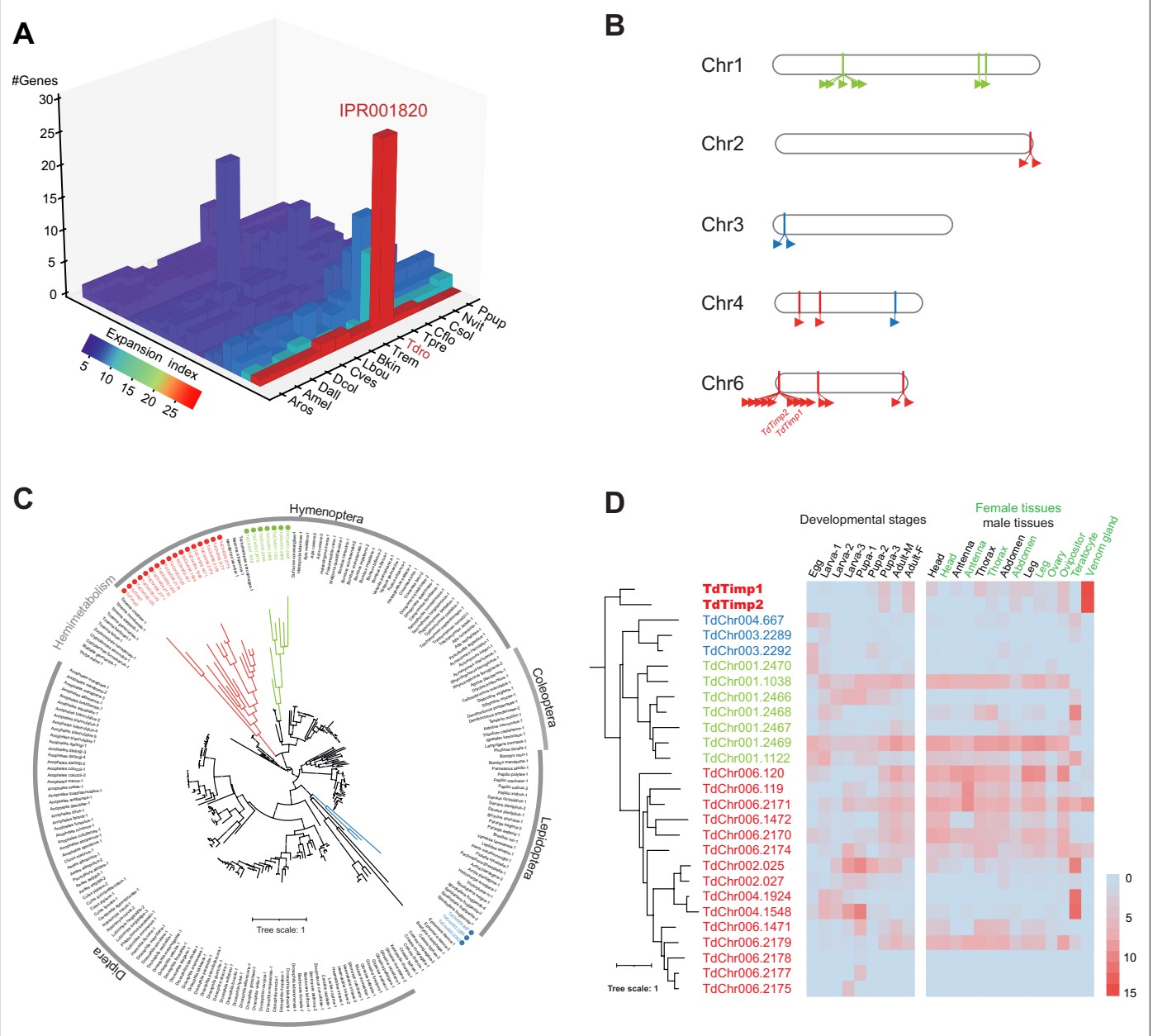

**Figure 2.** Genomic distribution and expression profiling of *Timps* in *Trichopria drosophilae* (Td). (**A**) Top 20 expanded functional domains in Td genome in comparison to other parasitoids. Species abbreviations correspond to those in *Figure 1D*. (**B**) Genomic locations of characterized metalloproteinases (*Timp*) genes on Td chromosomes. (**C**) Maximum-likelihood phylogenetic tree across all identified Timp in insects. The copies in Td are highlighted in colour. The colours also correspond to those on (**B**) and (**D**). (**D**) Transcriptome expression profiles of *Timp* genes across different developmental stages and representative tissues. The phylogeny is based on the expression pattern.

Orthologue analysis showed that half of each parasitoid gene set consisted of universal genes across species (*Figure 1—figure supplement 3*). Interestingly, approximately 83% of universal genes in Td experienced gene duplication events. By sorting all the expanded gene families within and between species, we found that the gene family encoding tissue inhibitors of metalloproteinases (TIMPs; IPR001820) was an extreme outlier (*Figure 2A*, *Supplementary file 2*). Td genome encodes 29 *Timp* genes, much more than that of any other hymenopteran species (0–7 genes). These copies were located in 10 neighbouring regions of all Td chromosomes and formed three phylogenetic clades unique to Td, indicating multiple recent duplications (*Figure 2B and C*). Expression profiling of *Timps* showed widespread expression in most copies except for a unique sublineage that showed

specialized expression in the venom glands (*Figure 2D*). These two copies (termed *TdTimp1* and *TdTimp2*) are described below.

## Td arrests host development via injection of venom

High-quality, fully annotated genomic resources allowed us to explore the molecular architecture of various adaptive traits of Td. Td is a typical pupal parasitoid that lays eggs exclusively in the pupal stage of hosts; adult wasps emerge from the host pupae after approximately 18 days at 25 °C (*Figure 1A* and *Figure 3A*). The pupal stage of the hosts is a non-feeding stage that normally lasts no longer than 5 days for *D. suzukii* and *D. melanogaster* at 25 °C. Thus, pupal parasitoid wasps should have evolved unique strategies to utilize limited nutritional resources effectively during a limited period. One such adaptation is the consequent developmental arrest of the host, which is commonly observed in parasitized hosts (*Rivers and Denlinger, 1995*; *Beckage and Gelman, 2004*). We monitored this process in real-time at 24 hr intervals after parasitization of 1-day-old host pupae of *D. suzukii*. Compared to non-parasitized pupae, parasitized pupae showed evident defects in eye pigment deposition and bristle formation (*Figure 3B*, *Figure 3—figure supplement 1A*), strongly indicative of developmental arrest. These phenotypic alterations started to occur on the second day post-parasitization (*Figure 3B*, *Figure 3—figure supplement 1A*), indicating that the development of the host pupae was immediately stopped after parasitization. We also performed parasitization experiments on another host species, *D. melanogaster*, for parallel comparisons and in-depth functional genetic investigations. Here, a similar pattern of developmental arrest was observed when *D. melanogaster* pupae were provided for parasitization (*Figure 3C*, *Figure 3—figure supplement 1B*).

The most likely effector resulting in host development arrest is the venom, given that parasitoid wasps always inject venom into hosts along with oviposition to condition and manipulate the host for the successful development of wasp progeny (*Werren et al., 2010*; *Asgari and Rivers, 2011*; *Moreau and Asgari, 2015*). To test this hypothesis, the venom fluid from the reservoir of a single Td female was isolated and injected into 1 day host pupae under two levels of dilution (1:20 and 1:40) to monitor host development (*Figure 3A*). As expected, doses 1:20 and 1:40 markedly delayed the development of both host species in comparison to the control (*Figure 3B and C*, *Figure 3—figure supplement 1*). Moreover, the alternating effects of a higher dose of venom (1:20 dilution) were very similar to those parasitized by Td and stronger than those of the lower dose (1:40 dilution). Thus, our results suggest that Td may arrest host development via the coinjection of venom.

## Timps are the main venom compounds responsible for host development arrest

To uncover the venom components that confer host developmental arrest, we sequenced the transcriptome of Td venom glands (VGs) and characterized 641 genes with solid expression evidence (*Supplementary file 3*). Given that not all expressed genes are necessarily translated and secreted into the reservoir, we performed LC–MS/MS on Td venom to further characterize reliable venom proteins (VPs) (*Supplementary file 3*). Of the 27 highly expressed (Z test, $p<0.05$) VPs with known functions, a pair of *Timp* genes with the highest expression values attracted the most attention (*Figure 3D*). Surprisingly, these two *Timp* genes were the previously highlighted *Timp1* and *Timp2* genes that showed expression specialization of venoms from the massive expansion in Td genome (*Figure 2*).

In insects, matrix metalloproteinases (Mmps) regulate pupal development by degrading extracellular matrix proteins, and Timp delays pupal development (*Jia et al., 2017*; *Jia and Li, 2023*). Thus, we wondered whether Td uses these specialized Timps to arrest host development. To test this hypothesis, we mimicked Td parasitization by applying a modified UAS/GAL4 system to drive spatiotemporally specific expression of exogenous *Timp* in *Drosophila* pupae (*Figure 3E*). As expected, the development of *D. melanogaster* pupae was markedly delayed upon the expression of *DmTimp* (*Figure 3F*, *Figure 3—figure supplements 2 and 3*). Although ectopic expression of either *TdTimp1* or *TdTimp2* did not lead to a visible developmental delay, driving the expression of both *TdTimp1* and *TdTimp2* arrested the development of *D. melanogaster* as effectively as the expression of *DmTimp* (*Figure 3F*, *Figure 3—figure supplements 2 and 3*). Taken together, these results suggest that cocktails of TdTimp1 and TdTimp2 are bioactive compounds of the venom that result in host development arrest.

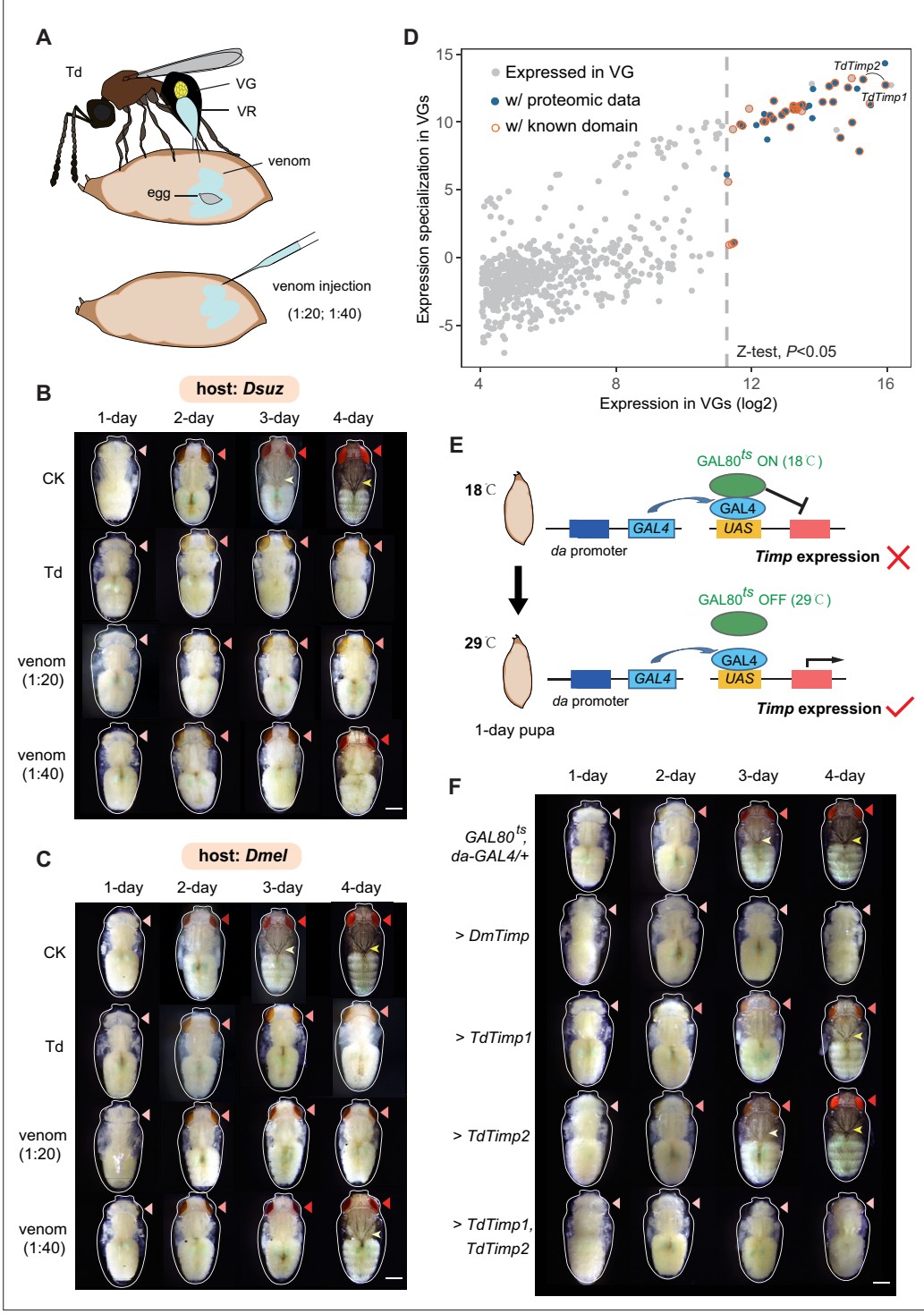

**Figure 3.** *Trichopria drosophilae* (Td) venom glands recruit a family of metalloproteinases (Timp) as venom proteins to arrest the host development. (**A**) Schematic diagram of Td parasitization and venom injection into *Drosophila* host pupa. Wasp venom is produced in the venom gland (VG) and stored in the venom reservoir (VR). (**B**) and (**C**) Development of *D. suzukii* (**B**) and *D. melanogaster* (**C**) pupae treated by Td parasitization and venom injections. One-day-old host pupae were either parasitized by Td or injected with wasp venom at 1:20 and 1:40 dilutions or PBS (CK). Fly eyes are marked by red arrowheads, and bristles are marked by yellow arrowheads. Here, darker red eye colour and bristle appearance represent the older development of host pupae. Scale bar: 400 μm. More than 40 host pupae were examined for each group. (**D**) Identification and annotation of venom proteins (VPs). The expression value in the VGs and its specialization level are presented on the x-axis and y-axis, respectively.

*Figure 3 continued on next page*

*Figure 3 continued*

The specialized index was estimated by the ratio of expression in VGs to the mean value of any other tissues and developmental stages (*Supplementary file 3*). (**E**) Schematic illustration of the temperature-sensitive conditional GAL4/GAL80$^{ts}$ system to drive *UAS-Timp* expression in host pupae. At the permissive temperature (18 °C), binding of GAL80$^{ts}$ to GAL4 prevented the transcription of *Timp* genes. After shifting to 29 °C, GAL80$^{ts}$ was inactivated, allowing *da-GAL4* to ubiquitously activate the transcription of *Timp* genes. (**F**) Development of *D. melanogaster* pupae after ectopic expression of *D. melanogaster Timp* (*DmTimp*) and *TdTimps*. One-day-old *D. melanogaster* pupae were transferred from 18 to 29°C using the GAL4/GAL80$^{ts}$ system to ubiquitously overexpress *Timp* genes in host pupae. Fly eyes are marked by red arrowheads, and bristles are marked by yellow arrowheads. Here, darker red eye colour and bristle appearance represent the older development of host pupae. The outline around each representative image refers to the pupa case. Scale bar: 400 µm. More than 40 host pupae were examined for each group.

The online version of this article includes the following figure supplement(s) for figure 3:

**Figure supplement 1.** Characterization of the developmental delay phenotype in *Figure 3B* (**A**) and *Figure 3C* (**B**).

**Figure supplement 2.** Development of *D. melanogaster* UAS-controls.

**Figure supplement 3.** Characterization of the developmental delay phenotype in *Figure 3F* and *Figure 3— figure supplement 2*.

## Td releases teratocytes to dissociate host tissues

Parasitoid wasp larvae feed largely on host fluids for development. We found that as host development was arrested, Td simultaneously released a special group of cells into the host haemocoel along with egg hatching (*Figure 4A*, *Figure 4—figure supplement 1*). Approximately 100 cells were secreted into the host haemocoel after larval hatching (*Figure 4—figure supplement 2A*). The cell number remained constant during the first three days, then decreased drastically, and finally vanished on day 8 (*Figure 4—figure supplement 2A*); that is, these cells no longer divided after being released. Moreover, these cells rapidly grew, with initially 85±13 µm in diameter and up to approximately 249±57 µm before vanishment (*Figure 4—figure supplement 2B* and C). These characteristics agree with those of teratocytes found in other parasitoids (*Hotta et al., 2001*; *Pedata et al., 2003*; *Firlej et al., 2007*). Thus, we defined these cells as teratocytes, which is the first discovery of teratocytes in the family Diapriidae. Transmission electron microscopy of Td teratocytes revealed a dense lawn of microvilli on the cell surface and that each cell consisted of a spherical nucleus, substantial amounts of rough endoplasmic reticulum, and mitochondria (*Figure 4B*). These features indicate high levels of protein synthesis and secretory activity in Td teratocytes.

We observed that host tissues started to be digested after the teratocytes were released into the host haemocoel. Do the teratocytes confer the digestion process? We tested this hypothesis by coculturing Td teratocytes and host tissues in vitro. Td teratocytes were purified from parasitized hosts and cultured with representative *D. suzukii* tissues, including the brain, testes, and ovaries, in Schneider's medium. Compared with the control, which was maintained for 30 hr without digestion, host tissues incubated with teratocytes from a single parasitoid were partially digested at 12 hr and fully digested after 30 hr (*Figure 4C*). These findings suggest that teratocytes facilitate the effective dissociation of host tissues, which may play a key role in nutritional exploitation for parasitoid larvae development.

## Teratocytes digest host tissues via trypsins

Although teratocytes have been reported in several other species, their molecular composition remains largely unknown in general (*Burke and Strand, 2014*; *Strand, 2014*). We further conducted transcriptomic profiling to characterize the teratocyte genes that facilitate host tissue digestion. A total of 3784 genes were expressed in teratocytes (*Supplementary file 4*). The majority of the 129 genes with significantly high expression (Z-test, $p<0.05$) were ribosome- (48 genes) and digestion-related genes (23 genes) (*Figure 4D*). Eight trypsin genes dominated among the highly expressed digestion-related genes in teratocytes, indicating their essential role in digesting host tissues.

Trypsin is the most abundant proteinase in invertebrate digestive systems (*Muhlia-Almazán et al., 2008*; *Dias et al., 2015*; *Shao et al., 2021*). To verify their functions in parasitization, trypsin inhibitors (TLCK and TPCK) were added to the incubation system of teratocytes and host tissues (*D. suzukii* testes) to determine whether digestion could be prevented. Along with the increase in

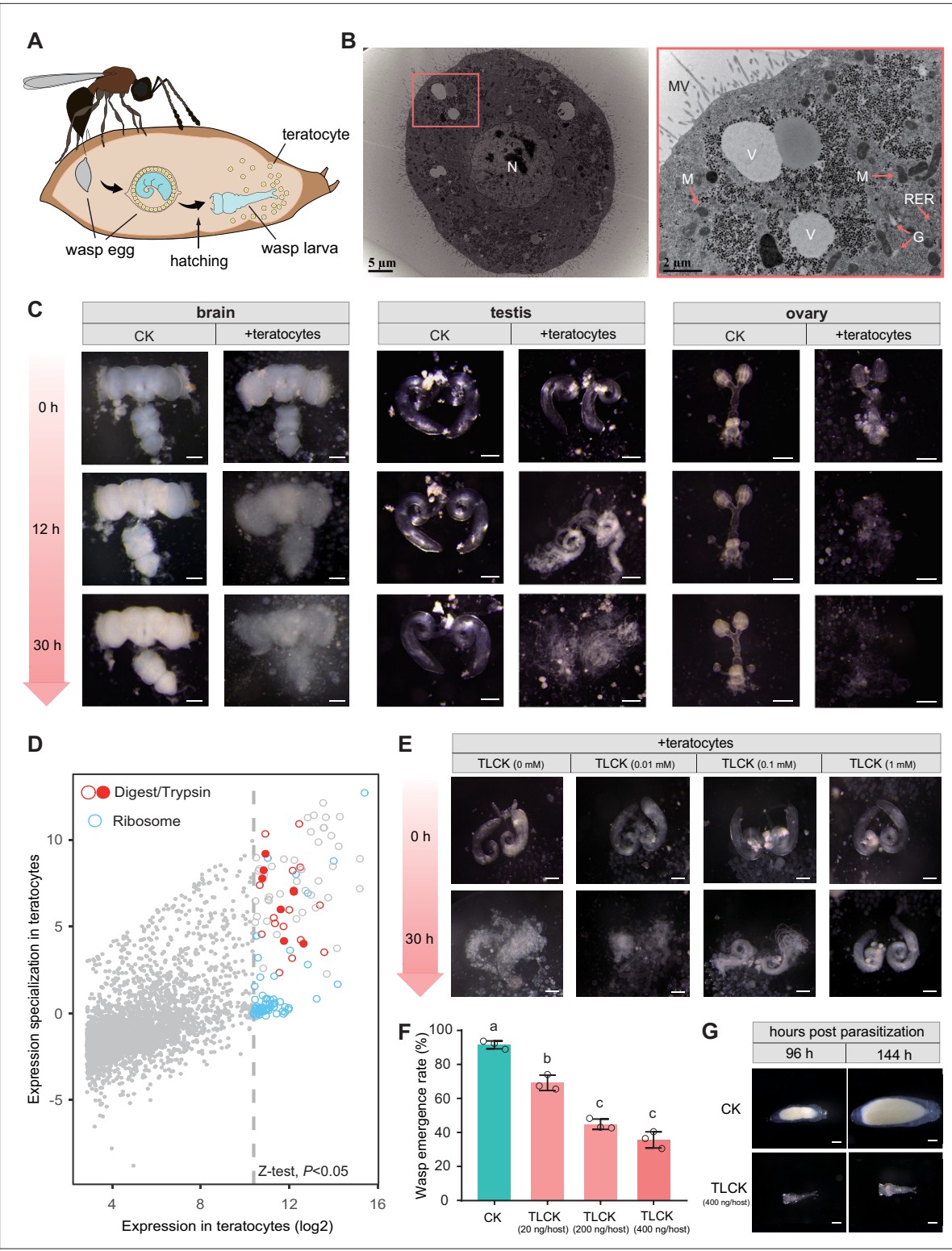

**Figure 4.** *Trichopria drosophilae* (Td) teratocytes secret trypsin proteins to digest host tissues. (**A**) Schematic diagram of teratocytes in Td-parasitized *Drosophila* hosts. Teratocytes are derived from the serosal membrane during Td egg hatching. (**B**) Transmission electron microscopy of Td teratocytes. Details of the micrograph (red box) showing the perinuclear region containing abundant rough endoplasmic reticulum (RER), numerous mitochondria (**M**), Golgi apparatus (**G**), and vesicles (**V**). Note that microvilli (MV) are obvious on the external surface of the teratocyte membrane, and the nucleus (**N**) is relatively large. Scale bars: shown in the images. (**C**) *D. suzukii* host tissues, including the brain, testis, and ovary, were cocultured with teratocytes derived from a single parasitized host. Host tissues cocultured with Schneider's medium only were used as controls (CK). The status of the host tissues

*Figure 4 continued on next page*

*Figure 4 continued*

was recorded after 0 hr, 12 hr, and 30 hr of incubation. Scale bars: 200 µm. More than 20 tissues were examined for each group. (**D**) Identification of effecting components in teratocytes. The expression value in the teratocytes and associated specialized level to other tissues and developmental stages are presented on the x-axis and y-axis, respectively. The specialized index was calculated as the ratio of expression value in teratocytes to the mean value of other tissues and developmental stages (***Supplementary file 4***). (**E**) Different concentrations of a trypsin inhibitor, TLCK, were added to the mixtures of *D. suzukii* testes and teratocytes. The status of the host tissues was recorded after 0 hr and 30 hr of incubation. Scale bars: 200 µm. More than 20 testes were examined for each group. (**F**) The wasp emergence rate in *D. suzukii* parasitized by Td after injection of different amounts of TLCK (20 ng, 200 ng, 400 ng per host pupa). ddH$_2$O injection was used as a control (CK). Three biological replicates were performed. Data represent the mean ± SEM. Significance was analysed by one-way ANOVA with Sidak's multiple comparisons test. Different letters indicate statistically significant differences (p<0.05). (**G**) Representative images of Td larvae in CK and TLCK-injected hosts after 96 hr and 144 hr parasitization. Scale bars: 200 µm. More than 20 larvae were examined for each group.

The online version of this article includes the following source data and figure supplement(s) for figure 4:

**Source data 1.** Related to *Figure 4F*.

**Figure supplement 1.** The teratocytes are released during *Trichopria drosophilae* (Td) egg hatching.

**Figure supplement 2.** The size and number of teratocytes in parasitized hosts.

**Figure supplement 2—source data 1.** Related to *Figure 4—figure supplement 2A*.

**Figure supplement 2—source data 2.** Related to *Figure 4—figure supplement 2B*.

**Figure supplement 3.** A trypsin inhibitor, TLCK, impairs the degradation function of teratocyte.

**Figure supplement 4.** A trypsin inhibitor, TPCK, impairs the degradation function of teratocyte.

**Figure supplement 4—source data 1.** Related to *Figure 4—figure supplement 4C*.

**Figure supplement 5.** Inhibition of trypsins impairs the development of wasp larvae.

**Figure supplement 5—source data 1.** Related to *Figure 4—figure supplement 5*.

inhibitor concentration, we found that the digestion of host tissues was gradually inhibited (***Figure 4E***, ***Figure 4—figure supplement 3***, ***Figure 4—figure supplement 4A*** and B). We then injected trypsin inhibitors directly into the parasitized host pupae to investigate whether this digestion process benefits parasitization. As expected, applying a trypsin inhibitor resulted in significantly lower emergence rates of parasitoid wasps, especially upon the injection of higher amounts of inhibitors (***Figure 4F***, ***Figure 4—figure supplement 4C***). Correspondingly, we noticed that the trypsin inhibitor injection led to abnormal wasp larvae development. Upon injection of 400 ng TLCK per host, the parasitoid larvae were found to be significantly smaller than the control on day 4 and almost dead on day 6 post-parasitization, owing to the lack of well-digested food (***Figure 4G***, ***Figure 4—figure supplement 5***). Altogether, these results suggest that teratocytes degrade host tissues using digestive enzymes, such as trypsin, to provide essential nutrients for the development and survival of wasps. Other digestion-related enzymes may also contribute to host tissue digestion, although their functions deserve further investigation.

## Td benefits from parasitizing young host pupae but cannot discriminate between young and old hosts

We showed that Td had evolved the functional integration of arresting host development and digesting host tissues to facilitate the utilization of limited resources in host pupae. In drosophilids, larval structures are lysed during the early pupal period, and adult structures begin to develop during the late period. Thus, Td should parasitize younger host pupae to earn longer developmental times and better nutritional resources. Indeed, the respective functions of venom and teratocytes were tested with the prerequisite of parasitization on newly pupated hosts (***Figure 3*** and ***Figure 4***). By testing the parasitism efficiency in host pupae of different ages (1- to 4-day-old *D. suzukii* or *D. melanogaster*), we found that Td fed old host pupae (4-day-old) showed a dramatic reduction in both parasitism rate and wasp emergence rate (p<0.001) (***Figure 5A and B***, ***Figure 5—figure supplement 1A*** and B). Simultaneously, both male and female wasps that emerged from young hosts (1-day-old) exhibited larger body sizes than those that emerged from old pupae (Student's t-test, p<0.001) (***Figure 5C***, ***Figure 5—figure supplement 1C***). For parasitoids, successful parasitism largely depends on the development of ovaries and VGs in females (***Segoli and Rosenheim, 2013***; ***Moreau and Asgari, 2015***; ***Huang et al., 2021***). We further found that the ovaries and VGs of females that emerged from young hosts were

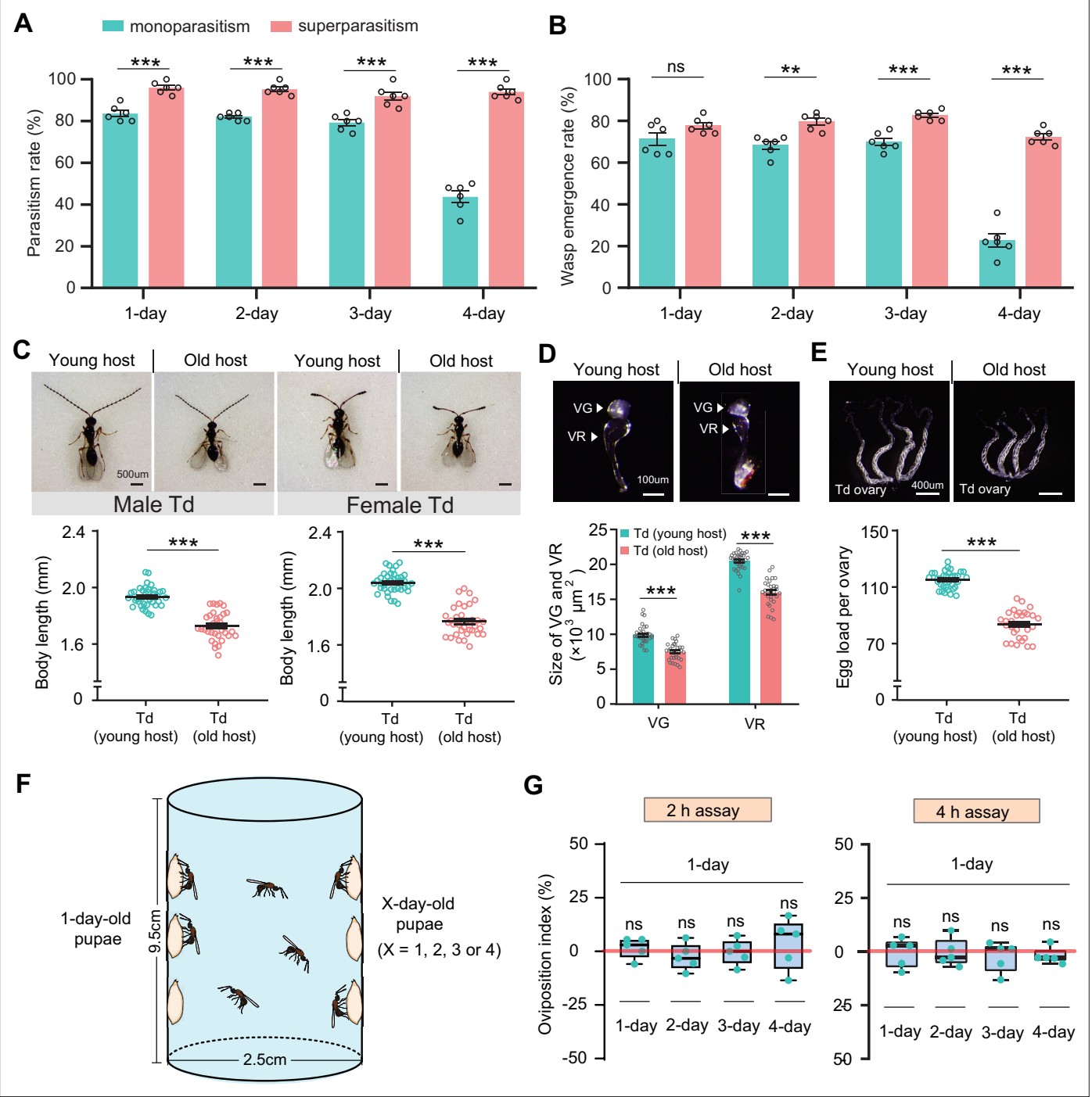

**Figure 5.** Superparasitism enhances *Trichopria drosophilae* (Td) parasitic efficiency. (**A**) and (**B**) Parasitism rates (**A**) and wasp emergence rates (**B**) in 1- to 4-day-old *D. suzukii* pupae after monoparasitism or superparasitism by Td. Six biological replicates were performed. Data represent the mean ± SEM. Significance was analysed by two-way ANOVA with Sidak's multiple comparisons test (ns, not significant; **p<0.01; ***p<0.001). (**C**) The images and body length of Td males and females emerged from young and old *D. suzukii* hosts. n=38 and 35 male wasps, n=39, and 34 female wasps, respectively. Data represent the mean ± SEM. Significance was determined by two-tailed unpaired Student's t-tests (***p<0.001). Scale bar: 500 µm. (**D**) The size of the venom gland (VG) and venom reservoir (VR) of Td females emerged from young and old *D. suzukii* hosts. Left to right: n=33, 31, 33, and 31 wasps. Data represent the mean ± SEM. Significance was determined by two-tailed unpaired Student's t-tests (***p<0.001). Scale bar: 100 µm. (**E**) The ovaries and egg load of Td females emerged from young and old *D. suzukii* hosts. n=33 and 30 wasps, respectively. Data represent the mean ± SEM. Significance was determined by two-tailed unpaired Student's t-tests (***p<0.001). Scale bar: 400 µm. (**F**) Schematic diagram of the Td oviposition choice assays. One-day-old host pupae were placed on the left side, and the same number of X-day-old host pupae (X=1, 2, 3, or 4) were placed on the right side. Oviposition index = $(N_1 − N_x)/(N_1 + N_x) \times 100\%$, where $N_1$ is the total number of wasp eggs in the host pupae on the left and $N_x$ is the total

*Figure 5 continued on next page*

*Figure 5 continued*

number of wasp eggs in the host pupae on the right. (**G**) Oviposition indices of Td females in different aged *D. suzukii* hosts from 2 hr and 4 hr assays. Five biological replicates were performed. Box plots represent the median (bold black line), quartiles (boxes), and minimum and maximum (whiskers). All oviposition indices were not significantly different from zero (bold red line), which represents no choice for Td egg laying. Deviation of the oviposition index against zero was tested with the Wilcoxon signed rank test (ns, not significant). Young host: 1-day-old pupae; old host: 4-day-old pupae.

The online version of this article includes the following source data and figure supplement(s) for figure 5:

**Source data 1.** Related to *Figure 5A*.

**Source data 2.** Related to *Figure 5B*.

**Source data 3.** Related to *Figure 5C*.

**Source data 4.** Related to *Figure 5D*.

**Source data 5.** Related to *Figure 5E*.

**Source data 6.** Related to *Figure 5G*.

**Figure supplement 1.** Superparasitism enhances *Trichopria drosophilae* (Td) parasitic efficiency on *D. melanogaster*.

**Figure supplement 1—source data 1.** Related to *Figure 5—figure supplement 1A*.

**Figure supplement 1—source data 2.** Related to *Figure 5—figure supplement 1B*.

**Figure supplement 1—source data 3.** Related to *Figure 5—figure supplement 1C*.

**Figure supplement 1—source data 4.** Related to *Figure 5—figure supplement 1D*.

**Figure supplement 1—source data 5.** Related to *Figure 5—figure supplement 1E*.

**Figure supplement 1—source data 6.** Related to *Figure 5—figure supplement 1F*.

**Figure supplement 2.** The image of the host parasitized by two *Trichopria drosophilae* (Td) female wasps.

significantly larger than those from old hosts (p<0.001) (*Figure 5D and E*, *Figure 5—figure supplement 1D* and E). By dissecting the ovaries, we found significantly more mature eggs from females that emerged from young host pupae (p<0.001, *Figure 5E*, *Figure 5—figure supplement 1E*). These results showed that the parasitization of young hosts benefits the fitness of Td.

The next compelling question was whether Td wasps could recognize host pupae of different ages before parasitization and subsequently prefer to parasitize younger pupae for maximum fitness. Thus, we designed a two-choice oviposition assay and monitored the host selection preferences of Td wasps (*Figure 5F*). Specifically, 20 young hosts (1-day-old *D. suzukii* or *D. melanogaster* pupae) were attached to the left side of a container, whereas pupae of different ages (1- to 4-day-old) were attached to the right side for competition. Ten Td female wasps (3-day-old) were released into the bottle and allowed to choose the host for oviposition freely (*Figure 5F*). After a certain time (2 or 4 hr for oviposition choice), the number of eggs laid in the host pupae between the two sides was compared to determine the oviposition preference. Regardless of whether the choice lasted for 2 or 4 hr, Td wasps showed no significant preference for either young or old host pupae (*Figure 5G*, *Figure 5—figure supplement 1F*). These results indicate that Td wasps have not evolved the ability to discriminate the young host pupae as a priority for parasitization.

## Superparasitism enhances parasitic success in older hosts

The parasitic success of Td largely relies on locating a young host; however, Td does not have the ability to discriminate between young and old hosts. Whether Td has evolved any adaptive strategies to compensate for this disadvantage? We noted that more than one female wasp parasitized the same host in both the laboratory and field (*Figure 5—figure supplement 2*), leading us to speculate that unexpected superparasitic behaviour (laying more than one egg per host) might be allowed or adaptive in Td. To test this hypothesis, we evaluated the parasitic efficiency of hosts with multiple infections. Strikingly, both the parasitism and emergence rates were significantly improved by superparasitism (*Figure 5A and B*). The resulting benefit was particularly evident in parasitizing old pupae (4-day-old), in which low parasitic efficiencies by monoparasitism were recovered 2.2-fold (parasitism rate) and 3.2-fold (emergence rate) by superparasitism (*Figure 5A and B*).

Superparasitism is generally uncommon in solitary parasitoids because only one adult can emerge from the host. Despite the better parasitization performance, we noted that only one adult Td emerged from the superparasitized pupae. Superparasitism likely leads to the additional introduction

of venom and teratocytes, which maximizes the utilization of host nutrition, especially in older hosts with poorer resources. Therefore, superparasitism in Td may partially compensate for the inability to recognize young pupae as hosts.

### Td avoids interspecific competition with larval parasitoids

Approximately 50 parasitoids have been reported to attack drosophilids (*Carton et al., 1986*), including larval (e.g. Aj) and pupal parasitoids (e.g. Td). This overlapping ecological niche leads to potential interspecific competition between larval and pupal parasitoids within the same host range. Because Td allows conditional intraspecific competition, the next compelling question would be whether Td allows interspecific competition with larval parasitoids. We provided Td with *D. suzukii* or *D. melanogaster* pupae that had been parasitized by Aj at the larval stage (*Figure 6A*). We found that Td wasps used their ovipositors to sting Aj-parasitized host pupae, similar to what they did for non-parasitized pupae. However, almost no eggs were laid inside these Aj-parasitized Td-stung pupae, in contrast to the high oviposition rate (approximately 93% for both *Drosophila* hosts) of non-Aj-parasitized pupae (*Figure 6B*). Thus, Td could discriminate between larva-parasitized hosts to avoid competition with larval parasitoids.

Given that Td wasps use ovipositors to sting both parasitized and non-parasitized pupae before oviposition, we hypothesised that Td ovipositors, in addition to the well-defined function of oviposition, might serve as sensory organs to evaluate the status of host pupae. Td ovipositor is an anatomical and functional cluster consisting of an ovipositor sheath and ovipositor stylet (*Figure 6C*), of which the unsheathed stylet is more likely to probe and stab. Using scanning electron microscopy, we found that the ovipositor stylet had two types of sensilla on its dorsal and ventral valves: the surface-dome and the coeloconic sensilla (*Figure 6D–I*). Coeloconic sensilla was the most abundant type, and secretory pores and surface-dome sensilla were also found on Td ovipositor stylets (*Figure 6F–I*). These sensilla and secretory pores occurred on the distal end of the ovipositor stylet but not on the proximal end, and there were more sensilla and secretory pores on the dorsal valve than on the ventral valve (*Figure 6D–F*). These anatomical features support a possible role of the ovipositor stylet in host discrimination.

We sequenced the transcriptome of the ovipositors and made a comprehensive comparison with various Td tissues and organs. Based on the overall expression profiles, principal component analysis placed the ovipositor closer to the antennae and legs of both males and females (*Figure 6J*), indicating a strong functional correlation among these anatomically non-contact organs. Given that the antennae and legs are well-defined sensory organs for volatile and contact semiochemicals in insects (*Vosshall et al., 1999*; *Schütz et al., 1999*; *Scott et al., 2001*; *Kim et al., 2003*), respectively, the expression correlation supports the role of oviposition in sensing host environmental signals. By further comparing the expression profiles of ovipositors and female antennae, we found that three chemosensory protein genes (CSPs) and two odourant-binding protein genes (OBPs) showed extremely high expression in both tissues, as well as a few genes encoding gustatory receptors (GRs) and olfactory receptors (ORs) (*Figure 6K*). More interestingly, the general co-receptor, Orco, which forms heteromultimeric complexes with other ligand-selective ORs, does not express in the ovipositor (*Figure 6—figure supplement 1*) and most genes related to odourant reception (i.e. OBPs and other ORs) showed biased expression in the antennae, while most taste recognition-related genes, i.e., GRs and CSPs, were more highly expressed in the ovipositor (*Figure 6K*). Correspondingly, we found a few GRs with the highest expression in ovipositors (*Figure 7*; *Supplementary file 5*). These patterns further support the hypothesis that, unlike the antennae that confer odourant chemoreception, Td ovipositors are more likely to serve as taste organs by sensing contact signals inside the host pupae for host discrimination. How gustation regulates the senses of other parasitoids requires further investigation.

## Discussion

### Td has evolved an integrated molecular adaptation for nutrition utilization

Parasitoid wasps are a highly diverse group of insects that are renowned for their diverse life history strategies and co-evolution with their hosts (*Pennacchio and Strand, 2006*; *Harvey et al., 2013*; *van*

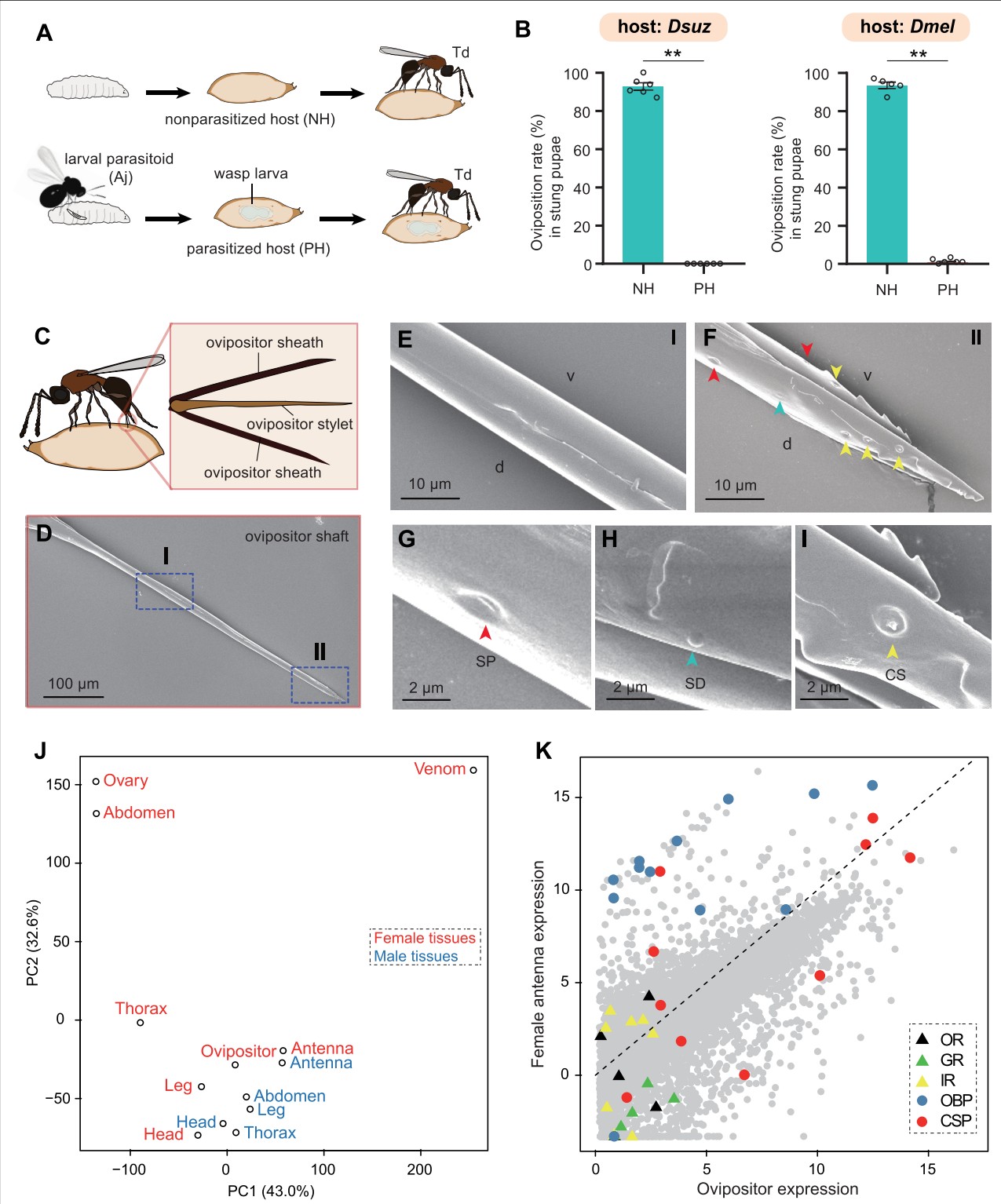

**Figure 6.** *Trichopria drosophilae* (Td) avoids interspecific competition through chemoreception in the ovipositor. (**A**) Schematic diagram of the Td interspecific competition assay. PH, Aj-parasitized host; NH, non-parasitized host. (**B**) Oviposition rates of Td in NH and PH pupae of *D. suzukii* and *D. melanogaster*, respectively. At least five biological replicates were performed. Data represent the mean ± SEM. Significance was analysed by the Mann-Whitney U test (**p<0.01). (**C**) Schematic diagram Td ovipositor, which contains two ovipositor sheaths and one ovipositor stylet. (**D**) to (**I**) Transmission electron microscope micrograph of Td ovipositor stylet. Higher magnification micrographs show the proximal end (**E**), Region I in (**D**) and distal end (**F**, II in **D**) of the ovipositor stylet. Different types of sensilla (arrowheads) on ventral (**v**) and dorsal (**d**) valves are identified, including secretary pores (SP, red

*Figure 6 continued on next page*

*Figure 6 continued*

arrowhead) (**G**), surface-dome sensilla (SD, green arrowhead) (**H**) and coeloconic sensilla (CS, yellow arrowhead) (**I**). Scale bars are shown in the images. (**J**) Principal component analysis based on overall expression profiles across different tissues. (**K**) Expression profiles of Td genes in ovipositors and female antennae. Each dot indicates the log-transformed expression values (TPM) in ovipositors (x-axis) and in female antennae (y-axis).

The online version of this article includes the following source data and figure supplement(s) for figure 6:

**Source data 1.** Related to *Figure 6B*.

**Figure supplement 1.** Transcriptome expression profile of *Orco* on female antenna and ovipositor of *Trichopria drosophilae* (Td).

de Kamp et al., 2018; Ode et al., 2022). Many koinobiont parasitoids that attack early host periods (e.g. larvae) maintain hosts alive and growing for sustained resource availability (**Brodeur and Boivin, 2004**; **Pennacchio and Strand, 2006**), while certain idiobiont parasitoids that usually attack non-feeding host stages (e.g. pupae) alternatively arrests host development for high-quality resources. *Drosophila* are natural hosts for multiple parasitic wasps of different genera, including both larval and pupal parasitoids. Their parasitic strategies have been widely studied and reported in multiple parasitoid species, with a particular focus on venom evolution and behavioural adaptations (**Mortimer et al., 2013**; **Martinson et al., 2017**; **Ramroop et al., 2021**; **Huang et al., 2021**; **Chen et al., 2021**; **Wertheim, 2022**). How to conflict with host immune responses and manipulate the host to create a suitable environment are important for larval parasitoids because the hosts are relatively active and proficient in immunity during the larval stage. For example, previous studies have shown that immune suppressors and virulence proteins dominate the VPs of *Leptopilina* larval parasitoids (**Colinet et al., 2011**; **Colinet et al., 2013**; **Poirié et al., 2014**; **Heavner et al., 2017**; **Huang et al., 2021**).

Td is a highly successful pupal parasitoid colonizing a broad range of *Drosophila* species, including *D. suzukii*. In the present study, we demonstrated the molecular adaptation of Td to achieve parasitic success (**Figure 8**). Td uses venom to arrest host pupal development and maintain young status via the recruitment of *Timps* (**Figure 3**). Moreover, Td released a special type of cell along with egg hatching, teratocytes, to accelerate the digestion of host tissues (**Figure 4**). The functional integration

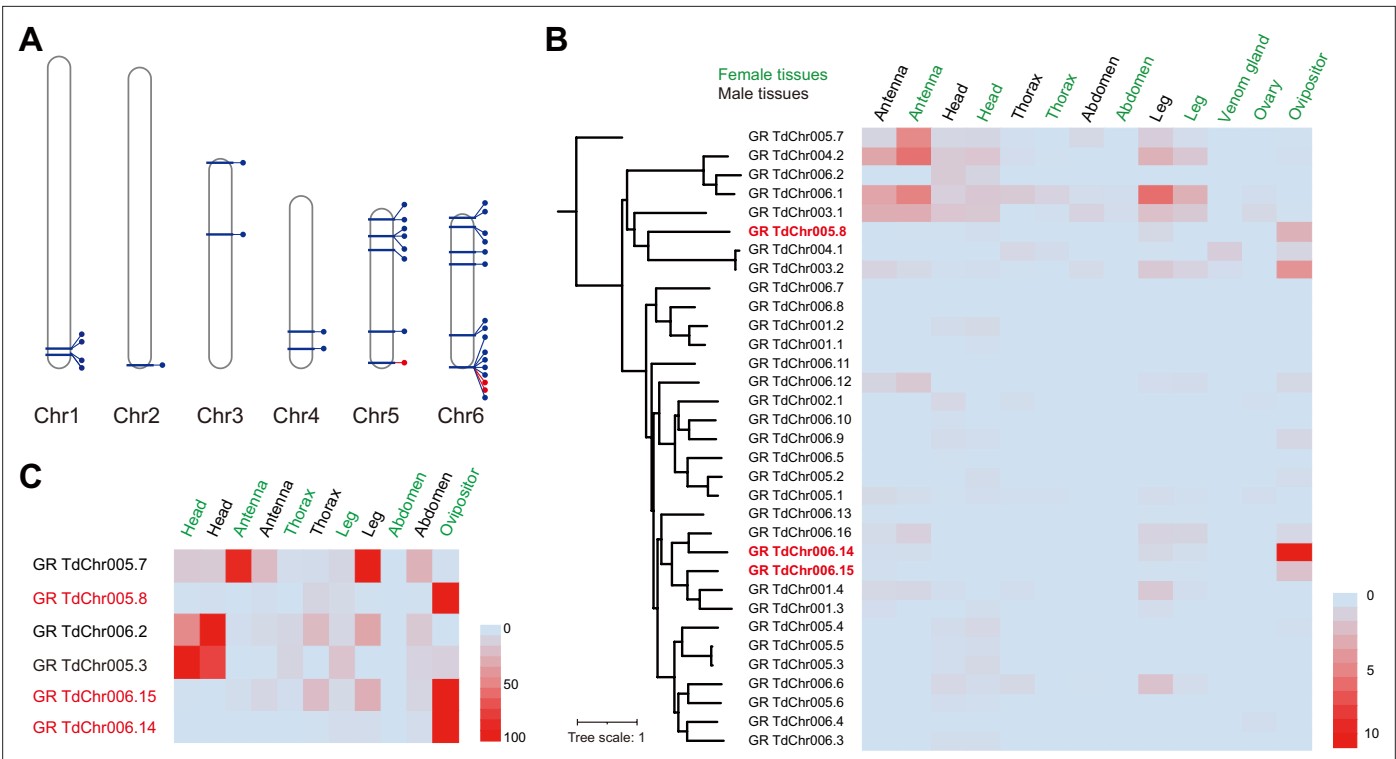

**Figure 7.** Genomic distribution and expression profiling of gustatory receptor genes in *Trichopria drosophilae* (Td). (**A**) Genomic location of characterized gustatory receptor (GR) genes on Td chromosomes. (**B**) Transcriptome expression profiles of GR genes across representative tissues. The phylogeny is based on the expression pattern. (**C**) qPCR verification of selective GRs with high expression in ovipositors.

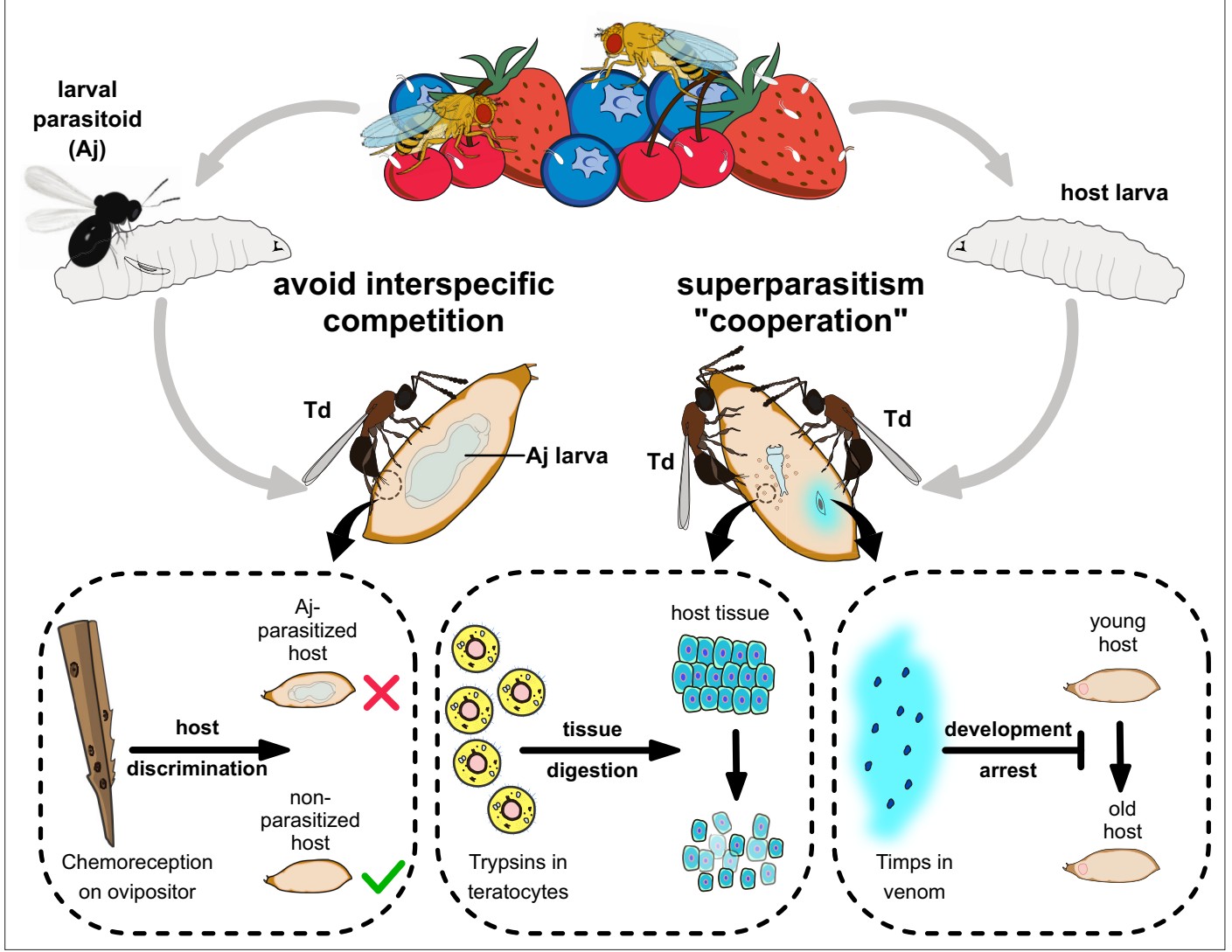

**Figure 8.** Proposed model of the molecular and ecological adaptions underlying the parasitic success in *Trichopria drosophilae* (Td). Td is a solitary parasitoid that presents high levels of pupal parasitism to *Drosophila* species, including *D. suzukii*. Td allows intraspecific superparatism to increase the success of wasp emergence from hosts with the help of two weapons. Specifically, two Timp venom proteins (VPs) trigger the development arrest of *Drosophila* hosts, eventually leading to the maintenance of the hosts at a younger status, whereas trypsins in teratocytes contribute to the digestion of host tissues, providing suitable nutritional resources for the wasp larvae. Moreover, Td avoids interspecific competition with larval parasitoid species to extend their parasitic efficacy and mostly depends on chemoreception on the ovipositor.

benefits Td as the utilization of nutritional resources is highly efficient during a limited period. The adaptations adopted by pupal parasitoids are largely unique to those of larval parasitoids and highly associated with the non-feeding and relatively inactive conditions of pupae. Correspondingly, few characterized genes of these parasitic effectors were related to immune responses (***Supplementary file 3 and 4***), indicating that conflict with host immunity plays a minor role in the adaptation of pupal parasitoids.

Another genus of pupal parasitoids, *Nasonia*, that feeds on multiple fly species but not drosophilid flies, has been the subject of genetic, ecological, developmental, and behavioural research for many years (***Beukeboom and Desplan, 2003***; ***Werren et al., 2009***; ***Werren et al., 2010***), but it serves as a model for specialized investigations of pupal parasitization. Notably, many strategies uncovered in this study have not been described in *Nasonia*, which may advance our understanding of the parasitism of pupae and the utilization of Td as a unique system for studying ecological niche differentiation of parasitoids.

## Gene duplication and expression shift play a dominant role in Td functional adaptations

Parasitic effects on organs and tissues provide excellent systems for studying the evolutionary modes of novel genes. Although it was proposed that the co-option of single-copy genes plays a common role in recruiting venom genes in certain parasitoids, such as *Nasonia* (*Martinson et al., 2017*), specific cases have also been reported highlighting gene duplication events, including duplicated *Lar* in the venom of a generalist *L. heterotoma* that suppresses host immune responses and the massive expansion of *Warm* in *L. boulardi*, which helps the specialist parasitoid evade host immune attacks (*Huang et al., 2021*). These differences suggest a highly diverse evolutionary mode of parasitism across different genera. In this study, we characterized Timps as a major compound in the venom that confers host developmental arrest (*Figure 3*), trypsin as a major component in teratocytes that facilitates host digestion (*Figure 4*), and a set of chemoreception proteins recruited by ovipositors for taste recognition (*Figure 6*). Strikingly, all affected genes were found to have undergone massive duplication events.

As described in the genome-wide analysis, *Timp* was the most expanded gene family in Td genome (*Figure 2A*). By searching for Timps across insects, we found that this gene family was mainly present in holometabolous insects such as Diptera, Lepidoptera, Coleoptera, and Hymenoptera (*Figure 2C*), possibly because of its role in regulating metamorphosis in this class of insects. The massive expansion of *Timps* in Td was derived from three independent phylogenetic sublineages, including one clustered with other hymenopteran copies, one clustered with patchy copies inside the clade of other insect orders, and one that is relatively unique (*Figure 2C*). The third sublineage is of the most expansion, in which most components were located on chromosome 6 and showed widespread expression across most tissues and developmental stages (*Figure 2B and D*). The two VP genes (*TdTimp1* and *TdTimp2*) that confer host development arrest were derived from tandem duplication of this sublineage and further experienced expression specialization in venom. We noted another four copies of this sublineage that possibly translocated to other chromosomes, showing expression specialization in larvae and teratocytes (*Figure 2B and D*). Similarly, we found that a few GRs originated from tandem duplications and experienced an expression shift to the ovipositor (*Figure 7*). Thus, we propose that gene expression, followed by gene expansion, may play a common role in the unique functional adaptations of Td.

## Td allows intraspecific competition but refuses interspecific competition from larval parasitoids

Parasitoid wasps are also excellent systems for studying the complex interrelationships within an ecosystem, because their survival and reproduction depend highly on the limited nutrients within a single host (*Harvey et al., 2013*; *Ode et al., 2022*). In addition to molecular adaptations, we uncovered ecological adaptations underlying Td parasitism, including the conditional allowance of intraspecific competition and strict refusal of interspecific competition (*Figure 8*).

Superparasitism usually lowers the reproductive efficiency of parasitoids by affecting their development, resulting in premature mortality of both hosts and parasitoids. Many parasitoid species avoid superparasitism by refusing to lay eggs in their parasitized hosts. For instance, the larval parasitoid *L. boulardi* utilizes EsGAP VPs to trigger parasitized hosts' escape behaviour, resulting in avoiding superparasitism (*Chen et al., 2021*). However, superparasitism is conditionally allowed if it benefits parasitoids. Despite the evolved molecular adaptations, we found that Td's parasitic success largely depended on its location to a host pupa of the appropriate age (newly pupated) (*Figure 5*). However, the ability to discriminate young pupae is unavailable in Td (*Figure 5G*). Superparasitism was frequently observed in Td and was further verified by its role in improving parasitic efficiency in old hosts (*Figure 5A and B*) despite the emergence of only one parasitoid. We hypothesized that the additional introduction of venom and teratocytes by partners within a single host pupa might accelerate the nutritional utilization of pupal parasitoids. Thus, intraspecific competition partially serves as a collaboration that can maximize the survival rate of offspring and favour the population at the cost of individuals.

Interspecific competition is widespread in the highly overlapped ecological niches among parasitoids. Approximately 50 hymenopteran parasitoids have been reported to infect various *Drosophila* species (*Carton et al., 1986*), including larval parasitoids from Figitidae (e.g. *Leptopilina*) and

Braconidae (e.g. Aj) as well as pupal parasitoids from Pteromalidae and Diapriidae (e.g. Td). Except for Pteromalidae parasitoids, most are highly specialized in *Drosophila*. Importantly, larval parasitoids have priority in occupying hosts, imposing strong selection pressure on pupal parasitoids to locate hosts. Unlike intraspecific competition, Td strictly avoided interspecific competition with larval parasitoids. Our results showed that Td effectively recognized and refused to lay eggs in Aj-parasitized hosts (*Figure 6*). Furthermore, this ability possibly depended on the evolved sensilla on the ovipositor stylet of Td, along with the ability for host discrimination via taste recognition (*Figure 6*). These uncovered ecological adaptations of Td provide valuable empirical knowledge regarding how organisms deal with inter- and intraspecific competition in nature and how ecological adaptations reconcile with functional innovations.

## Insights into the utilization of parasitoids as agents of pest control

In addition to serving as a study system in biology and ecology, parasitoids have been widely used as biological control agents because of their environmental friendliness and ability to suppress pest populations substantially (*Smith, 1996*; *Beckage and Gelman, 2004*; *Liu et al., 2015*). Our study began with identifying a highly successful parasitoid on *D. suzukii* and a broad range of other drosophilids. Although a few larval parasitoid species in Asia, such as Aj, have been reported to successfully parasitize *D. suzukii*, other parasitoids either do not prefer *D. suzukii* or show poor parasitic efficiency on *D. suzukii* (*Chabert et al., 2012*; *Mazzetto et al., 2016*; *Kremmer et al., 2017*). We estimated that the parasitic efficiency of Aj in *D. suzukii* in this study was approximately 50% (*Figure 1B and C*). In contrast, the great parasitic performance of Td on *D. suzukii* has been observed in this study, indicating the prospective potential of Td utility in the biological control of *D. suzukii*. The molecular and ecological adaptations of Td will provide important knowledge for further improving this potential agent.

# Materials and methods
## Insects

The strain of pupal parasitoid *T. drosophilae* (Td) trapped from orchards in Hangzhou, Zhejiang, China (30.29°N, 120.08°E) in May 2016 was used for experiments (strain Hangzhou), while the strain of larval parasitoid *A. japonica* (Aj) collected from orchards in Taizhou, Zhejiang, China (28.50°N, 120.34°E) in June 2018 was used for control parasitization assays (strain Taizhou). The complete mitochondrial genomes of these two strains were sequenced and submitted to GenBank with the accession numbers MN966974 (Td) and MN882556 (Aj), respectively. The primers for COI sequence amplification of the two wasp species were provided in *Supplementary file 6*. Both strains were developed from an individual mother and maintained on *D. melanogaster* strain *Canton-S* in the laboratory. Td was allowed to parasitize *Drosophila* pupae, and Aj was allowed to parasitize 2nd instar *Drosophila* larvae. The infected hosts were maintained at 25 °C until adult wasps emerged. The newly emerged wasps were collected into vials containing apple juice agar medium until exposure to hosts. The apple juice agar recipe was 27 g agar, 33 g brown sugar, and 330 ml pure apple juice in 1000 ml diluted water.

*D. melanogaster Canton-S* (#BL64349) and *UAS-Timp* (#BL58707) lines were acquired from the Bloomington *Drosophila* Stock Center. The ubiquitous driver *da-GAL4* (#TB00103) line was acquired from the Tsing Hua Fly Center. *D. simulans* (#BCF93), *D. pseudoobscura* (#BCF95) and *D. virilis* (#BCF97) were acquired from the Core Facility of *Drosophila* Resource and Technology, Chinese Academy of Sciences. *D. suzukii* (Strain Hangzhou) was collected in the above traps with their parasitoid wasps, and its mitochondrial genome accession number in GenBank is MW237837.1. *D. santomea* was provided by Dr. Qi Zhou (Zhejiang University, China).

To generate *UAS-TdTimp1* and *UAS-TdTimp2* transgenic flies, the coding regions of *TdTimp1* and *TdTimp2* were amplified from the cDNA of Td wasps using the primers listed in *Supplementary file 6*. The resulting cDNA fragment was digested with NotI (NEB, USA) and EcoRI (NEB, USA) and inserted into the pUAST-attB vector. The plasmids of *TdTimp1* and *TdTimp2* were injected into *Drosophila* embryos and integrated into the *attP* sites on chromosome 2 (cytological loci 25C6) and chromosome 3 (cytological loci 86F8), respectively.

## Genome sequencing

Td samples for genome sequencing have been maintained in the laboratory for at least 50 generations. To generate a high-quality chromosome-level genome, Td genome was fully sequenced using long-read sequencing technology (PacBio) and Hi-C. For PacBio sequencing, DNA was extracted from a pool of ~1000 male Td adults of a monoisolate to meet the requirements for library construction using a DNeasy Blood and Tissue Kit (Qiagen, Germany). DNA quality was checked using gel electrophoresis (0.8% agarose gel) and a High Sensitivity DNA Kit (Agilent, USA) in combination with a Bioanalyzer 2100 (Agilent, USA). A 20 Kb genomic library was constructed and sequenced by Berry Genomics Co., Ltd. (Beijing, China) according to the standard protocols on a PacBio Sequel platform. A total of 120.1 Gb raw reads were generated, accounting for 128 x genome coverage. This DNA sample was also sequenced using the Illumina platform to yield 57.7 Gb (~61.5 X coverage) short reads for the purpose of error correction.

Hi-C libraries were constructed from a pool of 20 newly emerged Td males according to the method previously described in *Lieberman-Aiden et al., 2009*. Briefly, the samples were fixed with 2% formaldehyde for 10 min at room temperature. Then, glycine solution was added to a final concentration of 100 mM to stop cross-linking. The cross-linked DNA was extracted and digested with HindIII (NEB, USA) overnight. Biotin-14-dCTP17 was introduced during the sticky end repair process. The interacting DNA fragments were ligated to form chimeric junctions using T4 DNA ligase. Finally, Hi-C library sequencing was performed by GrandOmics Co., Ltd. (Wuhan, China) on an Illumina HiseqX Ten platform (Illumina, HiseqX Ten, USA), generating a total of 143.7 Gb paired-end reads (2×150 bp), accounting for 153 x coverage.

## Chromosome staining

Chromosome preparation of Td wasp for karyology was carried out according to the method by Imai et al. with some modifications (*Imai et al., 1988*). Briefly, the cerebral ganglia of both male and female Td early pupae were dissected in Ringer's saline solution under a stereoscope (Leica, Germany) and washed for 3–5 min in Ringer's buffer. The cerebral ganglia were then incubated with 0.005% colchicine-hypotonic solution (diluted in 1% sodium citrate) for 1 hr and transferred onto a clean slide. The tissues were fixed in fixative solution A (anhydrous ethanol: glacial acetic acid: double-distilled water at 3:3:4 by volume), then the samples were gently broken up with forceps to fully distribute the chromosomes. Finally, the samples were fixed in fixative solution B (anhydrous ethanol: glacial acetic acid at 1:1 by volume) and mounted in ProLong Gold Antifade Mountant with DAPI (Invitrogen, USA). Fluorescence images were captured on an LSM 800 confocal microscope (Zeiss, Germany).

## Genome assembly

PacBio reads were used to assemble contigs using Nextdenovo v2.4.0 (*Hu et al., 2024*) with the parameters 'read_type = clr read_cutof = 2 k genome_size = 900 m seed_depth = 60 nextgraph_options=-a 1 A.' Assembled contigs were corrected and polished using high coverage of Illumina paired-end reads using Nextpolish v1.3.1 (*Hu et al., 2020*) with the following parameters: 'task = best rerun = 3 sgs_options=-max_depth 100 -bwa lgs_options=-min_read_len 1 k -max_depth 100 lgs_minimap2_options=-x map-pb.' The polished contigs were then subjected to removal of redundancy using purge_dups v1.2.3 (*Guan et al., 2020*) for two rounds with default parameters.

To realize a chromosomal-level assembly, assembled contigs were further clustered based on Hi-C contact information from Hi-C sequencing reads. Juicer v1.5.7 (*Capurso et al., 2020*) was used to process the contact signals that were provided for 3d-dna v190716 for chromosome grouping with the parameters '-q 1 --editor-repeat-coverage 2'.

The completeness of the genome assembly was estimated using the BUSCO pipeline v5.4.3 (*Waterhouse et al., 2018*; *Seppey et al., 2019*) against insect_odb10.

## Transcriptome sequencing and analysis

Td transcriptomes of different developmental stages were obtained from *Zhou et al., 2019*, including egg, L1 (Days 1–2 larvae, early larval stage), L2 (Days 3–4 larvae, middle larval stage), L3 (Days 5–6 larvae, late larval stage), P1 (Days 1–3 pupae, early pupal stage), P2 (Days 4–6 pupae, middle pupal stage), P3 (Days 7–9 pupae, late pupal stage), AM (Day 1 male adults), and AF (Day 1 female adults). The other Td samples for transcriptome analysis included 4-day-old teratocytes, VGs of Day 3–5 AF

wasps, and the different parts of Day 3–5 AF and AM Td wasps, including antenna, head without antenna, legs, thorax without legs, ovipositor, and abdomen without ovipositor. All samples were dissected in Ringer's saline solution on an ice plate under a stereoscope (Leica, Germany). Total RNA was independently extracted from each sample using the RNeasy MiniKit (Qiagen, Germany). Construction of the cDNA library and paired-end RNAseq (Illumina, NovaSeq 6000, USA) were carried out by Berry Genomics Co. Ltd. (Beijing, China). Statistics of transcriptome sequencing data are listed in *Supplementary file 7*. Each sample was independently mapped to the reference genome using HISAT2 v2.0.5 (*Kim et al., 2015*) using default parameters. The mapping results were processed using SAMTOOLS v1.9 (*Li et al., 2009*) as inputs for genome annotation.

We also sequenced full-length transcripts using the PacBio sequencing system (Pacific Biosciences, USA) for genome annotation. A total of 67.4 Gb transcriptome sequencing data of Td were generated from a library with an insert of 1–10 kb for the messenger RNA (mRNA) pool of all stages of Td development (*Supplementary file 7*). Raw reads were processed using isoseq v3.2.2 and mapped to the reference genome using minimap v2.1 (*Li, 2018*) with parameters of '-ax splice -uf --secondary=no C5.'

## Genome annotation

Repeat contents were annotated using the pipeline of RepeatMasker (https://www.repeatmasker.org/RepeatMasker/). First, a Td-specific repeat library was generated using RepeatModeler v2.0.2 (https://www.repeatmasker.org/RepeatMasker/). RepeatMasker v4.1.1 was then used to mask the repeat contents across the whole genome against both the Td-specific library and Dfam v3.2 (*Storer et al., 2021*). Protein-coding genes were predicted based on repeat-masked genome. Multiple gene prediction approaches were used to generate several gene sets, including (1) BRAKER v2.1.5 (*Brůna et al., 2021*) was applied to predict two gene sets on respectively transcriptome-based hints and related proteins-based hints; (2) Maker v2.31.10 (*Holt and Yandell, 2011*) was used to predict a gene set by calling SNAP v2006-07-28 (*Korf, 2004*) and Augustus v3.3.2 (*Stanke et al., 2008*) and integrating evidence of related proteins and full-length transcripts; (3) StringTie v2.0 (*Pertea et al., 2015*) was used to combine all Illumina-based transcriptome data to generate a merged transcript set using default parameters; (4) the python module 'collapse_isoforms_by_sam' of TOFU (https://github.com/Magdoll/cDNA_Cupcake) (*cDNA_Cupcake collaborators, 2022*) was used to generate a full-length reads-based transcript set, with parameters of '--dun-merge-5-shorter -c 0.9 -i 0.9.' All above independent gene sets were subject for pairwise comparisons at both transcript-level and exon-level and retained the consistent ones with the first priority and the ones with multiple lines of supports as the secondary priority. The ones with only support from a single set were excluded. The predicted genes were then annotated with functions based on both BLASTP v2.2.26 against NR databases and a local InterProScan search (v5.38–76.0) (*Jones et al., 2014*) for domains. Expression values of each gene in each developmental stage or tissue were estimated as TPM using salmon v0.12.0 (*Patro et al., 2017*) with parameters of 'quant -l A.' The expression of a specific gene in a given sample was determined based on the threshold value of N99, i.e., the value accounting for 99% of summarized transcripts in the corresponding sample.

## Evolutionary analyses

Orthologs across 14 hymenopteran species were sorted using OrthoFinder v2.5.4 (*Emms and Kelly, 2019*) with parameters '-S blast -M msa.' Orthologs with strict single copy in all 14 species were isolated to infer the phylogeny across species and concatenated to a super gene for each species. MAFFT v7.470 (*Katoh et al., 2002*) was used to perform multiple alignments and extracted with conserved regions by trimAI v1.4.rev15 (*Capella-Gutiérrez et al., 2009*). The phylogenetic tree was inferred using RAxML v8.2.12 (*Stamatakis, 2014*) with parameters '-f a -x 12,345 p 12345 -# 100 m PROTGAMMAJTTF.' The tree was illustrated and annotated using iTOL v6.6 (*Letunic and Bork, 2021*).

*Timp* homologs in other species were retrieved from the database of InterPro (https://www.ebi.ac.uk/interpro/) based on the domain IPR001820. All 198 insect genes encoding this domain were retrieved and used for phylogenetic analyses as described above.

## Venom preparation and microinjection

The venom reservoirs of Td female wasps aged 3–5 days were dissected in Ringer's saline solution on an ice plate under a stereoscope (Leica, Germany). The venom reservoirs were washed at least

three times in Ringer's buffer and placed into a 500 μl screw-top tube. The tube was then mixed with grinding glass beads with a size of 0.5 mm (Tansoole, China) and shaken on a vortex mixer six times for 10 s each time. After centrifugation at 10000×g at 4 °C for 5 min, the supernatant (venom fluid) was collected and quantified with the BCA Protein Assay Kit (Invitrogen, USA). Approximately 1:20 (16.2 ng) or 1:40 (8.1 ng) amounts of the total venom extract derived from a single female wasp were injected into each *D. melanogaster* and *D. suzukii* host pupa using the Eppendorf FemtoJet 4i device with the following parameters: injection pressure = 900 hPa; injection time = 0.15 s.

## Identification of Td venom proteins

For LC–MS/MS experiments, venom fluid from approximately 2000 Td female wasps aged 3–5 days was dissolved in 100 μl SDT lysis buffer (4% SDS, 100 mM Tris-HCl, 1 mM DTT, pH 7.6). The sample was boiled for 15 min and then centrifuged at 13,000×g at 4 °C for 40 min. The detergent and DTT were removed with repeated ultrafiltration (Microcon units) using UA buffer (8 M urea, 150 mM Tris-HCl, pH 8.0). Then 100 μl 50 mM iodoacetamide was added to block reduced cysteine residues, and the samples were incubated for 30 min in the dark. Then, the protein suspensions were digested with 3 μg trypsin (Promega, USA) in 40 μl 100 mM NH₄HCO3 buffer overnight at 37 °C, and the resulting peptides were desalted on C18 cartridges (Sigma, Germany), concentrated by vacuum centrifugation, and reconstituted in 40 μL of 0.1% (v/v) formic acid solution.

LC-MS/MS analysis was performed on a Q Exactive mass spectrometer (Thermo Fisher Scientific, USA) coupled to an Easy nLC HPLC liquid system (Thermo Fisher Scientific, USA). Briefly, 5 μg of the peptide mixture was loaded onto a reverse-phase trap column (Thermo Fisher Scientific Acclaim PepMap 100, 100 μm×2 cm, nanoViper C18) connected to the C18 reversed-phase analytical column (Thermo Fisher Scientific Easy Column, 10 cm long, 75 μm inner diameter, 3 μm resin) in buffer A (0.1% formic acid) and separated with a linear gradient of buffer B (84% acetonitrile and 0.1% formic acid) at a flow rate of 300 nl/min. The eluted peptides were ionized, and the full MS spectrum (from m/z 300–1800) was acquired by precursor ion scan using the Orbitrap analyser with a resolution of $r$=70,000 at m/z 200, followed by 20 MS/MS events in the Orbitrap analyser with a resolution of $r$=17,500 at m/z 200. The MS raw files were translated into mgf files and searched against the transcriptome of Td venom gland using Mascot 2.2 (*Perkins et al., 1999*). MS/MS tolerance was set at 20 ppm, and trypsin was defined as the cleavage enzyme allowing no more than two missed cleavages. Carbamidomethylation of cysteine was specified as a fixed modification, and oxidation of methionine was specified as a variable modification.

Venom-protein genes were finally defined based on both transcriptomic and proteomic evidence as described above. Genes with TPM higher than 16.55 (N99 value) in the venom were defined as venom gland-expressed genes. VG-expressed genes that could be fully aligned to at least three proteomic peptides were defined as venom proteins (*Supplementary file 3*). A Z test was performed to define significantly high expression (p<0.05).

## Ectopic overexpression of *Timps* in host pupae

To investigate whether ectopic expression of *DmTimp* and *TdTimp* could impair host development, we used a temperature-sensitive conditional GAL4/GAL80ts system to drive UAS-Timp expression only in host pupae (*Suster et al., 2004*; *Caygill and Brand, 2016*). Briefly, flies with different transgenic elements were raised at 18 °C to suppress da-GAL4 (a ubiquitous expression driver) activity during egg and larva developmental stages. After pupation, the 1 day *Drosophila* pupae were switched to 29 °C to inactive GAL80ts activity and thereby permit GAL4 activation of DmTimp and TdTimp, respectively.

## Host developmental status observation

The development of *D. melanogaster* and *D. suzukii* pupae was monitored using a stereoscope (Olympus MVX10, Japan) with a digital microscope camera (Olympus Dp47, Japan), and photos were taken at 24 hr intervals. The flies were removed from their pupal case and their eye colours were divided into five levels from white (young) to red (old), including white, light orange, orange, light red, and red. The number of flies with different eye colours was counted to measure the developmental effect. All these animals were pupae (after head eversion). The outline around each representative image in *Figure 3B, C and F*, and *Figure 3—figure supplement 2* refers to the pupa case.

## Parasitoid body, venom gland, and venom reservoir size measurement

Images of the whole body, venom gland, and venom reservoir of newly emerged Td from male pupae of *D. melanogaster* and *D. suzukii* of different ages were taken under a microscope (Olympus SZX16, Japan) with a digital camera (FluoCa Scientific, BioHD-C16, Singapore). The length and sizes of the venom gland and venom reservoir were measured by using ScopeImage9.0 software (Bioimager, Switzerland).

## Teratocyte number and size measurement

After parasitization by Td, the *Drosophila* hosts at different days post parasitization were dissected in Ringer's saline solution on an ice plate under a stereoscope (Leica, Germany). The number of teratocytes in each host was directly counted. In addition, teratocytes were imaged under a microscope (Olympus SZX16, Japan) with a digital camera (FluoCa Scientific, BioHD-C16, Singapore). The diameters of teratocytes were measured by using ScopeImage9.0 software (Bioimager, Switzerland).

## Scanning and transmission electron microscope analyses

For transmission electron microscopy analyses, Td teratocytes were fixed with 2.5% glutaraldehyde in phosphate-buffered saline (PBS, 0.1 M, pH 7.2) for 24 hr at 4 °C. Samples were washed three times in PBS for 15 min each and then postfixed in 1% OsO4 for 2 hr at 4 °C. Then, the samples were washed three times in PBS for 15 min each and dehydrated through a graded series of ethanol (30%, 50%, 70%, 80%, 90%, and 95% for 15 min each; 100% for 20 min each) and acetone (100%) twice for 20 min. Next, the samples were infiltrated successively with a graded mixture of acetone and Spurr resin (1:1 for 1 hr, 1:3 for 3 hr) and Spurr resin (SPI-Chem, USA) for 12 hr at room temperature. Finally, the samples were embedded in Spurr resin and polymerized at 70 °C for 12 hr. Ultrathin sections (~70 nm) were cut on an ultramicrotome (Leica EM UC7, Germany). The sections were stained with uranyl acetate and lead citrate and were observed and photographed using a transmission electron microscope (Hitachi, H-7650, Japan) operated at 80 kV.

For scanning electron microscope analyses, Td ovipositors were fixed with 2.5% glutaraldehyde in PBS for 24 hr at 4 °C. Samples were washed three times in PBS for 15 min each. Then, samples were postfixed with 1% OsO4 in PBS for 2 hr and washed three times in PBS for 15 min each. The fixed samples were dehydrated through a graded series of ethanol (30%, 50%, 70%, 80%, 90%, and 95%) for 15 min each and were then placed in absolute (100%) alcohol for 20 min. Samples were vacuum-dried and coated with gold-palladium using an Ion Sputter (Hitachi, E-1010, Japan) and observed with a scanning electron microscope (Hitachi, SU-8010, Japan) operated at 3 kV.

## Coculture of teratocytes with host tissues

Teratocytes were dissected from each host 4 days post parasitization and were collected in cell culture dishes containing 3 μl Schneider's *Drosophila* medium (Thermo Fisher Scientific, USA) plus antibiotics (ampicillin and kanamycin, each at a concentration of 100 μg/l). Then, the brain and the pair of testes and ovaries of *D. suzukii* host pupae were dissected. One single tissue was cocultured with teratocytes derived from a single host in Schneider's medium at room temperature. The status of the host tissues was recorded by a stereoscope (Olympus MVX10, Japan) with a digital microscope camera (Olympus Dp47, Japan) after 0 hr, 12 hr, and 30 hr of incubation.

## Coculture of trypsin inhibitors in vitro

Pairs of *D. suzukii* testes were dissected and individually cocultured with teratocytes derived from a single host in Schneider's medium. Two trypsin inhibitors, TLCK (tosyl-L-lysyl-chloromethane hydrochloride) (Sigma, Germany), were added at different concentrations of 0 mM (CK), 0.01 mM, 0.1 mM, and 1 mM; TPCK (L-chloromethyl (2-phenyl-1-(p-toluenesulfonylamino) ethyl) ketone) (Sigma, Germany) was added at different concentrations of 0 mM (CK), 0.005 mM, 0.05 mM, and 0.5 mM. The status of the host tissues was recorded by a stereoscope (Olympus MVX10, Japan) with a digital microscope camera (Olympus Dp47, Japan) after 0 hr and 30 hr of incubation.

## Microinjection of trypsin inhibitors in vivo

One-day-old *D. suzukii* pupae were parasitized by Td, and different amounts of TLCK (20 ng, 200 ng, and 400 ng) and TPCK (10 ng, 100 ng, and 200 ng) were injected into each host 48 hr post-infection.

We first used tweezers to make a hole at the abdomen of host pupal case, and the microinjection was done through the holes. The microinjection was conducted using the Eppendorf FemtoJet 4i device with the following parameters: injection pressure = 900 hPa; injection time = 0.15 s. Based on the volume of *D. suzukii* pupa, the approximate concentrations of TLCK were 0.04 mM (20 ng), 0.4 mM (200 ng), and 0.8 mM (400 ng), and the approximate concentrations of TPCK were 0.02 mM (10 ng), 0.2 mM (100 ng), and 0.4 mM (200 ng). The developmental status of wasp larvae was observed in the inhibitor-injected hosts 96 hr and 144 hr post-infection.

## Parasitic efficiency assay

For parasitic efficiency assays in *Figure 1B and C*, 3-day-old mated Td females were allowed to parasitize host pupae of different *Drosophila* species at a parasite/host ratio of 1:10 for 3 hr. In addition, 3-day-old mated Aj females were allowed to parasitize second instar host larvae of different *Drosophila* species at a parasite/host ratio of 1:10 for 2 hr. The parasitized hosts were maintained at 25 °C for analysis of parasitism efficiency.

For parasitic efficiency assays in *Figure 5A and B*, *Figure 5—figure supplement 1* A and B, Td females were allowed to parasitize different aged (e.g. 1 day, 2 day, 3 day, and 4-day-old) pupae of *D. melanogaster* and *D. suzukii*. We then selected hosts with monoparasitism and superparasitism for analysis of parasitism efficiency. Here, monoparasitism indicates that the host pupae were parasitized by Td one time; superparasitism indicates that the host pupae were parasitized by Td ≥2 times.

The parasitism rate and wasp emergence rate were calculated using the following formulas:

Parasitism rate = (1 − number of emerged host adults/number of total hosts)×100%; Wasp emergence rate = (number of emerged wasps/number of total hosts)×100%.

## Oviposition choice assay

Oviposition choice assays were carried out in a 2.5×9.5 cm cylindrical tube (*Figure 5F*). Briefly, twenty 1-day-old host pupae were sticky on the left side of the tube, and the same number of X-day-old host pupae (X=1, 2, 3, or 4) were sticky on the right side. Then, ten 3-day-old Td females were released into the apparatus for performing a 2 hr or 4 hr egg-laying choice assay. The tested hosts were dissected, and the total numbers of wasp eggs on both sides were counted. The oviposition preference indices (%) were calculated as $(N_1 − N_x)/(N_1 + N_x)×100\%$, where $N_1$ is the total number of wasp eggs in host pupae on the left side and $N_x$ is the total number of wasp eggs in host pupae on the right.

## Interspecific competition assay

Three-day-old Aj female wasps were allowed to parasitize 2nd instar larvae of *D. melanogaster* and *D. suzukii*, respectively. The infected hosts were maintained at 25 °C until pupation. Then, Aj-parasitized host pupae and regular pupae were provided to 3-day-old Td female wasps. Immediately, Td females used their ovipositor to sting the host pupae. Approximately thirty stabbed Aj-parasitized or regular host pupae were randomly selected and dissected under a microscope, and the portion of the oviposition rate of Td was calculated as the percentage that contained Td wasp egg(s) relative to the total number of hosts that were dissected. At least five replicates were performed for each group.

## Quantitative real-time PCR

Total RNA was extracted from different Td tissues (e.g. head, antenna, and so on) using the RNeasy Mini Kit (Qiagen, Germany) and then reverse transcribed into cDNA using HiScript III RT SuperMix for qPCR (Vazyme, China) following the manufacturer's protocol. qRT-PCR was performed in the AriaMx real-time PCR system (Agilent Technologies, USA) with the ChamQ SYBR qPCR Master Mix Kit (Vazyme, China) using the following temperature cycling conditions: 30 s at 95 °C, followed by 45 cycles of three-step PCR for 10 s at 95 °C, 20 s at 55 °C, and 20 s at 72 °C. The RNA levels of the target genes were normalized to that of *tubulin* mRNA, and the relative concentration was determined using the $2^{-\Delta\Delta Ct}$ method. All primers used for qRT-PCR in this study are listed in *Supplementary file 6*.

## Statistics

All statistical analyses were performed in GraphPad Prism version 8.0 (GraphPad Software, USA) and SPSS 26 (IBM, USA). Normal distribution of the data was tested using the Shapiro-Wilk test. The Bartlett chi-square test was used to test the homogeneity of variance of the data, which was consistent

with the normal distribution. We used two-tailed unpaired Student's t-tests to determine the statistical significance of a difference between two treatments when a parametric test was appropriate. We used the Mann-Whitney U test and Wilcoxon signed rank test for experiments requiring a nonparametric statistical test. ANOVA with Sidak's multiple comparisons tests were used to compare mean differences between multiple groups when a parametric test was appropriate. Fisher's exact test was used to compare the proportion of red eye host pupae and host tissue full digestion. Details of the statistical analysis are provided in the figure legends, including how significance was defined and the statistical methods used. Data represent the mean ± standard error of the mean (SEM). Different letters indicate statistically significant differences ($p<0.05$). For all other tests, significance values are indicated as * $p<0.05$; **$p<0.01$; ***$p<0.001$.

## Acknowledgements

We thank Bloomington *Drosophila* Stock Center, Tsing Hua Fly Center, Shanghai Core Facility of *Drosophila* Resource and Technology, and Dr. Qi Zhou for providing *Drosophila* stocks. We also thank Bio-ultrastructure analysis Lab of Analysis center of Agrobiology and environmental sciences, Zhejiang University for providing Scanning and transmission electron microscope sample preparation and observation platform. This study was supported by the Natural Science Foundation of China (Grants # 32325044, 32021001, 32302426, 32202375, 32225008), Chinese Academy of Sciences (XDB27040205 and XDPB16), and the Zhejiang Provincial Natural Science Foundation of China (LZ23C140003).

## Additional information

### Funding

| Funder | Grant reference number | Author |
|---|---|---|
| National Natural Science Foundation of China | 32325044 | Jianhua Huang |
| National Natural Science Foundation of China | 32021001 | Shuai Zhan |
| National Natural Science Foundation of China | 32302426 | Lan Pang |
| National Natural Science Foundation of China | 32202375 | Jiani Chen |
| National Natural Science Foundation of China | 32225008 | Shuai Zhan |
| Chinese Academy of Sciences | XDB27040205 | Shuai Zhan |
| Chinese Academy of Sciences | XDPB16 | Shuai Zhan |
| Zhejiang Provincial Natural Science Foundation of China | LZ23C140003 | Jianhua Huang |

The funders had no role in study design, data collection and interpretation, or the decision to submit the work for publication.

### Author contributions

Lan Pang, Gangqi Fang, Resources, Data curation, Formal analysis, Investigation, Visualization, Writing – original draft, Project administration; Zhiguo Liu, Zhi Dong, Resources, Formal analysis; Jiani Chen, Visualization; Ting Feng, Qichao Zhang, Investigation; Yifeng Sheng, Yueqi Lu, Ying Wang, Methodology; Yixiang Zhang, Guiyun Li, Resources; Xuexin Chen, Conceptualization; Shuai Zhan, Jianhua Huang, Conceptualization, Supervision, Funding acquisition, Writing – original draft, Writing - review and editing

## Author ORCIDs

Lan Pang http://orcid.org/0000-0003-1763-3392
Gangqi Fang http://orcid.org/0000-0003-4546-061X
Zhi Dong http://orcid.org/0000-0002-3179-5614
Jianhua Huang http://orcid.org/0000-0001-6509-1765

Reviewer #1 (Public review): https://doi.org/10.7554/eLife.94748.3.sa1
Reviewer #2 (Public review): https://doi.org/10.7554/eLife.94748.3.sa2
Author response https://doi.org/10.7554/eLife.94748.3.sa3

## Additional files

### Supplementary files

- Supplementary file 1. Basic features of *Trichopria drosophilae* genome.
- Supplementary file 2. Top 20 expanded gene families in *Trichopria drosophilae* (Td).
- Supplementary file 3. List of *Trichopria drosophilae* (Td) venom protein genes.
- Supplementary file 4. List of *Trichopria drosophilae* (Td) teratocyte genes.
- Supplementary file 5. List of *Trichopria drosophilae* (Td) chemoreception genes.
- Supplementary file 6. Primer sequences in this study.
- Supplementary file 7. Transcriptome sequencing data in this study.
- Supplementary file 8. Raw data in *Figure 1—figure supplement 1D*.
- MDAR checklist

### Data availability

The genomic data and associated transcriptome data are available in NCBI GenBank under BioProject numbers PRJNA922326. The MS proteome data were deposited to the ProteomeXchange Consortium (https://proteomecentral.proteomexchange.org) through the PRIDE partner repository (https://www.ebi.ac.uk/pride/) with the dataset identifier PXD038251. Other data generated or analysed during this study are included in the manuscript and supporting files.

The following datasets were generated:

| Author(s) | Year | Dataset title | Dataset URL | Database and Identifier |
|---|---|---|---|---|
| Lan P, Gangqi F | 2023 | The reference genome of *Trichopria Drosophilae* | https://www.ncbi.nlm.nih.gov/bioproject/PRJNA922326 | NCBI BioProject, PRJNA922326 |
| Gangqi F | 2024 | *Trichopria drosophilae* venom, LC-MS/MS-based proteomics | https://www.ebi.ac.uk/pride/archive/projects/PXD038251 | PRIDE, PXD038251 |

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
