## [Editor Report · eLife assessment]

The paper presents **valuable** insights into the success of the parasitoid *Trichopria drosophilae* on *Drosophila suzukii*, elucidating the importance of both molecular adaptations, such as specialized venom proteins and unique cell types, ecological strategies, including tolerance of intraspecific competition and avoidance of interspecific competition. Through **convincing** methodological approaches, the authors demonstrate how these adaptations optimize nutrient uptake and enhance parasitic success, highlighting the intricate coordination between molecular and ecological factors in driving parasitization success.

---

## [Referee Report · Reviewer #1 (Public review)]

Summary:

Major findings or outcomes include a genome for the wasp, characterization of the venom constituents and teratocyte and ovipositor expression profiles, as well as information about Trichopria ecology and parasitism strategies. It was found that Trichopria cannot discriminate among hosts by age, but can identify previously parasitized hosts. The authors also investigated whether superparasitism by Trichopria wasps improved parasitism outcomes (it did), presumably by increasing venom and teratocyte concentrations/densities. Elegant use of *Drosophila* ectopic expression tools allowed for functional characterization of venom components (Timps), and showed that these proteins are responsible for parasitoid-induced delays in host development. After finding that teratocytes produce a large number of proteases, experiments showed that these contribute to digestion of host tissues for parasite consumption.

The discussion ties these elements together by suggesting that genes used for aiding in parasitism via different parts of the parasitism arsenal arise from gene duplication and shifts in tissue of expression (to venom glands or teratocytes).

Strengths:

The strength of this manuscript is that it describes the parasitism strategies used by Trichopria wasps at a molecular and behavioral level with broad strokes. It represents a large amount of work that in previous decades might have been published in several different papers. Including all of these data in a manuscript together makes for a comprehensive and interesting study.

Weaknesses:

The weakness is that the breadth of the study results in fairly shallow mechanistic or functional results for any given facet of Trichopria's biology. Although none of the findings are especially novel given results from other parasitoid species in previous publications, integrating results together provides significant information about Trichopria biology.

---

## [Referee Report · Reviewer #2 (Public review)]

Summary:

Key findings of this research include the sequencing of the wasp's genome, identification of venom constituents and teratocytes, and examination of Trichopria *drosophilae* (Td)'s ecology and parasitic strategies. It was observed that Td doesn't distinguish between hosts based on age but can recognize previously parasitized hosts. The study also explored whether multiple parasitisms by Td improved outcomes, which indeed it did, possibly by increasing venom and teratocyte levels. Utilizing *Drosophila* ectopic expression tools, the authors functionally characterized venom components, specifically tissue inhibitors of metalloproteinases (Timps), which were found to cause delays in host development. Additionally, experiments revealed that teratocytes produce numerous proteases, aiding in the digestion of host tissues for parasite consumption. The discussion suggests that genes involved in different aspects of parasitism may arise from gene duplication and shifts in tissue expression to venom glands or teratocytes.

Strengths:

This manuscript provides an in-depth and detailed depiction of the parasitic strategies employed by Td wasps, spanning both molecular and behavioral aspects. It consolidates a significant amount of research that, in the past, might have been distributed across multiple papers. By presenting all this data in a single manuscript, it delivers a comprehensive and engaging study that could help future developments in the field of biological control against a major insect pest.

Weaknesses:

While none of the findings are particularly groundbreaking, as similar results have been reported for other parasitoid species in prior research, the integration of these results into one comprehensive overview offers valuable biological insights into an interesting new potential biocontrol species.

---

## [Author Response]

The following is the authors’ response to the original reviews.

**Public Reviews:**

**Reviewer #1 (Public Review):**
Summary:Major findings or outcomes include a genome for the wasp, characterization of the venom constituents and teratocyte and ovipositor expression profiles, as well as information about Trichopria ecology and parasitism strategies. It was found that Trichopria cannot discriminate among hosts by age, but can identify previously parasitized hosts. The authors also investigated whether superparasitism by Trichopria wasps improved parasitism outcomes (it did), presumably by increasing venom and teratocyte concentrations/densities. Elegant use of *Drosophila* ectopic expression tools allowed for functional characterization of venom components (Timps), and showed that these proteins are responsible for parasitoid-induced delays in host development. After finding that teratocytes produce a large number of proteases, experiments showed that these contribute to digestion of host tissues for parasite consumption.The discussion ties these elements together by suggesting that genes used for aiding in parasitism via different parts of the parasitism arsenal arise from gene duplication and shifts in tissue of expression (to venom glands or teratocytes).Strengths:The strength of this manuscript is that it describes the parasitism strategies used by Trichopria wasps at a molecular and behavioral level with broad strokes. It represents a large amount of work that in previous decades might have been published in several different papers. Including all of these data in a manuscript together makes for a comprehensive and interesting study.Weaknesses:The weakness is that the breadth of the study results in fairly shallow mechanistic or functional results for any given facet of Trichopria's biology. Although none of the findings are especially novel given results from other parasitoid species in previous publications, integrating results together provides significant information about Trichopria biology.

We thank the reviewer for appreciating the importance of our study.

**Reviewer #2 (Public Review):**
Summary:Key findings of this research include the sequencing of the wasp's genome, identification of venom constituents and teratocytes, and examination of Trichopria drosophilae (Td)'s ecology and parasitic strategies. It was observed that Td doesn't distinguish between hosts based on age but can recognize previously parasitized hosts. The study also explored whether multiple parasitisms by Td improved outcomes, which indeed it did, possibly by increasing venom and teratocyte levels. Utilizing *Drosophila* ectopic expression tools, the authors functionally characterized venom components, specifically tissue inhibitors of metalloproteinases (Timps), which were found to cause delays in host development. Additionally, experiments revealed that teratocytes produce numerous proteases, aiding in the digestion of host tissues for parasite consumption. The discussion suggests that genes involved in different aspects of parasitism may arise from gene duplication and shifts in tissue expression to venom glands or teratocytes.Strengths:This manuscript provides an in-depth and detailed depiction of the parasitic strategies employed by Td wasps, spanning both molecular and behavioral aspects. It consolidates a significant amount of research that, in the past, might have been distributed across multiple papers. By presenting all this data in a single manuscript, it delivers a comprehensive and engaging study that could help future developments in the field of biological control against a major insect pest.Weaknesses:While none of the findings are particularly groundbreaking, as similar results have been reported for other parasitoid species in prior research, the integration of these results into one comprehensive overview offers valuable biological insights into an interesting new potential biocontrol species.

We thank the reviewer for appreciating the importance of our study and for the suggestions on how to improve it.

**Reviewer #1 (Recommendations For The Authors):**
No additional comments
**Reviewer #2 (Recommendations For The Authors):**
Minor comments:Line 68 : would be better to spell out the name of the genus at first mention of the species

It has been corrected as suggested.

Lines 90-92 : This statement does to coincide with the figure. Could you please explain this better?

We have carefully checked the statement and the corresponding figure panels, but failed to find the disparity between them. Perhaps, the similar and neighboring labels of Dsuz and Dsan might cause confusion of the emergence rates. To further avoid this potential, we have modified fig.1b and 1c by highlighting the focal host Dsuz.

Lines 124: could you tell the mention of these genes (Piwi) is important in this context, particularly, for non- full-on experts in this field?

A previous study has revealed the relationship between the expansion of piwi and large genome, we meant to report a different pattern in our focal genome. We understand your confusion might be caused by the inserted statement regarding the repeat that separated them. Thus, we have moved the citation of previous finding to the place immediately precedent to the conclusion.

Line 233: "...composition remains largely unknown.." for Td or in general? Not clear..

Thank you. To make it clear, we have modified this sentence as “Although teratocytes have been reported in several other parasitoids, their molecular composition remains largely unknown in general”.

Line 286: "at a certain time".. confusing, please rephrase.

We have rephrased it as “After a certain time (2 or 4 hours for oviposition choice)”.

Line 293-294: I find this sentence quite hard to follow. Could you please rephrase it and/or expand this concept to make it clearer?

We have modified this sentence as “The parasitic success of Td largely relies on locating a young host; however, Td does not have the ability to discriminate between young and old hosts. Whether Td has evolved any adaptive strategies to compensate for this disadvantage?”

Line 314: "it would be interesting".. this is too weak of an argument. Please corroborate your motivation more soundly.

We have changed this statement as “Because Td allows conditional intraspecific competition, the next compelling question would be whether Td allows interspecific competition with larval parasitoids.”

Line 391: Divergent evolution is too of a big word in this context. I would tune it down to something like: "Studying ecological niche differentiation ".

Thank you. It has been corrected as suggested.